# Replication defective viral genomes exploit a cellular pro-survival mechanism to establish paramyxovirus persistence

Jie Xu[1], Yan Sun [1], Yize Li [2], Gordon Ruthel[1], Susan R. Weiss [2], Arjun Raj[3], Daniel Beiting[1] & Carolina B. López [1]

Replication defective viral genomes (DVGs) generated during virus replication are the primary triggers of antiviral immunity in many RNA virus infections. However, DVGs can also facilitate viral persistence. Why and how these two opposing functions of DVGs are achieved remain unknown. Here we report that during Sendai and respiratory syncytial virus infections DVGs selectively protect a subpopulation of cells from death, thereby promoting the establishment of persistent infections. We find that during Sendai virus infection this phenotype results from DVGs stimulating a mitochondrial antiviral-signaling (MAVS)-mediated TNF response that drives apoptosis of highly infected cells while extending the survival of cells enriched in DVGs. The pro-survival effect of TNF depends on the activity of the TNFR2/TRAF1 pathway that is regulated by MAVS signaling. These results identify TNF as a pivotal factor in determining cell fate during a viral infection and delineate a MAVS/TNFR2-mediated mechanism that drives the persistence of otherwise acute viruses.

[1] Department of Pathobiology, School of Veterinary Medicine, University of Pennsylvania, Philadelphia, PA 19104, USA. [2] Department of Microbiology, Perelman School of Medicine, University of Pennsylvania, Philadelphia, PA 19104, USA. [3] Department of Bioengineering, School of Engineering, University of Pennsylvania, Philadelphia, PA 19104, USA. Jie Xu and Yan Sun contributed equally to this work. Correspondence and requests for materials should be addressed to C.B.L. (email: lopezca@upenn.edu)

Persistent viral genomes are observed after a number of acute viral infections in humans, including respiratory syncytial virus (RSV), measles, and Ebola[1–3]. A number of host factors, such as impaired or altered cytokine production and progressive loss of immunological functions, support the maintenance of persistent infections[4]. However, the processes and cellular mechanisms determining the onset of viral persistence after acute viral infections remain unknown.

The innate immune response is the first active host barrier to virus replication and is essential to control the infection and activate adaptive responses that result in virus clearance. The antiviral innate response is initiated upon recognition of viral molecular patterns by cellular sensor molecules. Activation of these sensor pathways leads to the expression of genes with pro-inflammatory, antiviral, and pro-apoptotic activities that control virus growth and spread. During infections with important human pathogens including RSV, parainfluenza virus, and measles virus, the antiviral response is triggered by replication defective copy-back viral genomes (DVGs) that accumulate during viral replication[5–8]. DVGs potently stimulate intracellular RIG-I-like receptors (RLRs) that signal through the mitochondrial antiviral-signaling (MAVS) protein to stimulate the expression of genes that control virus replication and spread, and direct clearance of infected cells[9, 10].

Paradoxically, some species of DVGs can promote the establishment of persistent RSV, parainfluenza virus, measles virus, and other viruses during infections in tissue culture[11–14] and are proposed to be responsible for establishing persistent Ebola virus infections in humans[1]. This pro-persistence activity of DVGs has been related to the continuous competition for the viral polymerase between full-length genomes and DVGs, resulting in alternating cycles of replication of full-length and defective genomes[15–17]. However, this mechanism cannot explain the survival of virus-infected cells in the presence of strong pro-apoptotic and antiviral molecules, including type I IFNs and TNFα, that are induced in response to sensing of DVGs[10].

In order to better understand the host–virus interactions driving the establishment of persistent infections of otherwise acute viruses, we developed a technology that allowed us to investigate at a single cell level the mechanisms behind the different activities of DVGs in infected cells. Using fluorescent in situ hybridization targeting ribonucleic acid molecules (RNA FISH) to distinguish DVGs from standard viral genomes during infection, we reveal that during infection with the murine parainfluenza virus Sendai (SeV) or RSV DVGs accumulate only in a subpopulation of infected cells, and that these cells survive the infection longer than cells enriched in full-length virus Survival of DVG-high cells is dependent on MAVS signaling, and we identify TNFα produced in response to MAVS signaling as pivotal in determining cell fate during SeV infection. We show that while cells harboring full-length viral genomes die from virus-induced TNF-mediated apoptosis, cells enriched in DVGs regulate the expression and activity of a TNFR2/TRAF1 pro-survival program that protects them from TNF-induced apoptosis. Overall, this study reveals a mechanism by which distinct viral genomic products determine cell fate upon infection by taking advantage of the dual functions of TNFα to perpetuate both virus and host.

## Results

**DVGs dominate in a subpopulation of infected cells.** To better understand the impact of DVGs during infection, we established a RNA FISH assay that allowed us to differentiate SeV full-length genomes (FL-gSeV) from SeV DVGs at a single cell level. As copy-back DVGs are generated from the 5′ end of the viral genome and thus have a high sequence homology with the FL-gSeV[18, 19], we designed a two-color probing strategy to distinguish DVGs from FL genomes within infected cells (Fig. 1a). To detect replicating virus, a set of probes labeled with Quasar-570 (pseudo-colored red) was prepared against the 5′ end of the positive sense viral RNA and a different set of probes labeled with Quasar-670 (pseudo-colored green) was prepared against the 3′ end of the positive sense SeV genome, which covers the viral genomic sequence shared with DVGs. As a result of this design, DVGs are only bound by Quasar-670-labeled probes (denoted "DVG"), while FL-gSeV are bound by a combination of Quasar-570 and Quasar-670-labeled probes (denoted "FL-gSeV" and appearing as orange in the images) (Fig. 1a). To test the specificity of the labeling, we infected cells with SeV lacking DVGs (LD) alone or complemented with purified defective particles containing DVGs (pDPs). As expected, cells infected with SeV LD demonstrated a strong FL-gSeV signal and no DVG signal, while addition of pDPs resulted in detection of cells with strong positive DVG signal (Fig. 1b). Confirming their specificity, Quasar-670-labeled DVG probes detected in vitro transcribed DVG RNA transfected into cells and did not cross-react with the synthetic RNA poly I:C or with host GAPDH mRNA (Fig. 1c and Supplementary Fig. 1a, b).

Analysis of various epithelial cell lines infected with SeV containing high levels of DVGs (HD) showed an unexpected heterogeneous distribution of Quasar-570 and 670 signals at 24 and 48 h post infection. Differential accumulation of DVGs and FL-gSeV among infected cells (Fig. 1d and Supplementary Fig. 1c) occurred regardless of the initial amount of input virus (Fig. 1e and Supplementary Fig. 1d). The phenotype was recapitulated upon hybridization with probes that targeted the negative sense FL-gSeV (Supplementary Fig. 1e). Importantly, both FL-gSeV and DVGs probe pools had comparable sensitivity, which was similar to that of viral product-specific RT-qPCR (Supplementary Fig. 1f). Further, RT-qPCR of sorted cells negative for both Quasar-570 and 670 staining (Non-detected: ND, gated in blue), DVG-high (gated in green), and FL-gSeV high (FL-high, gated in orange), confirmed their differential content of FL-gSeV and DVG by RT-qPCR (Fig. 1f–h). Note that DVG-high cells also contained a relatively small amount of full-length viral genomes, which are expected to provide the viral machinery for DVG replication. An intermediate cell population showing strong Quasar-670 signal and positive for Quasar-570 (INT, gated in yellow) was also observed by flow cytometry (Fig. 1f, g). This population demonstrated moderate levels of SeV NP transcript, moderate FL-gSeV, and relatively high levels of DVGs by RT-qPCR (Fig. 1h). Because of its intermediate nature, we excluded this population from further analysis in this study.

To characterize the temporal distribution of DVGs among the infected cell population, we analyzed cells infected with SeV HD by RNA FISH followed by flow cytometry. FL-high and DVG-high cells were detected as early as 6 h post infection and their percentage increased during the time course of infection, corresponding well with cell imaging (Fig. 1i, j). DVG-high cells accumulated quickly upon infection, plateaued, and decreased at later time points, while accumulation of FL-high cells followed behind, reminiscent of the waves of accumulation of defective interfering particles and FL-genomes observed in long term infected cultures (Fig. 1k)[20, 21]. As expected based on their previous characterization as primary triggers of antiviral immunity[8, 10, 22], early accumulation of DVG-high cells was associated with expression of high levels of IFNB1 mRNA (Fig. 1l). In contrast, the kinetics of SeV NP mRNA expression followed that of the FL-high populations, but not that of the DVG-high population. Taken together, these data demonstrate a heterogeneous physical and functional distribution of DVGs and FL-gSeV during SeV infection.

**DVG-high cells survive infection and promote persistence**. To examine how DVG accumulation in a subpopulation of cells impacts the establishment of persistently infected cultures, we infected susceptible LLC-MK2 cells with SeV LD or HD and monitored the presence of infected cells overtime. Cultures infected with SeV LD died 8 days after infection (Fig. 2a, b), while

cultures infected with SeV HD recovered from a cell death crisis and were passaged for at least 17 days (Fig. 2a, b and Supplementary Fig. 2a). Remarkably, a substantially higher proportion of DVG-high cells survived by day 5 post SeV HD infection compared to FL-high cells (Fig. 2a), and by day 17 post infection ~34% of the total survivors were DVG-high compared

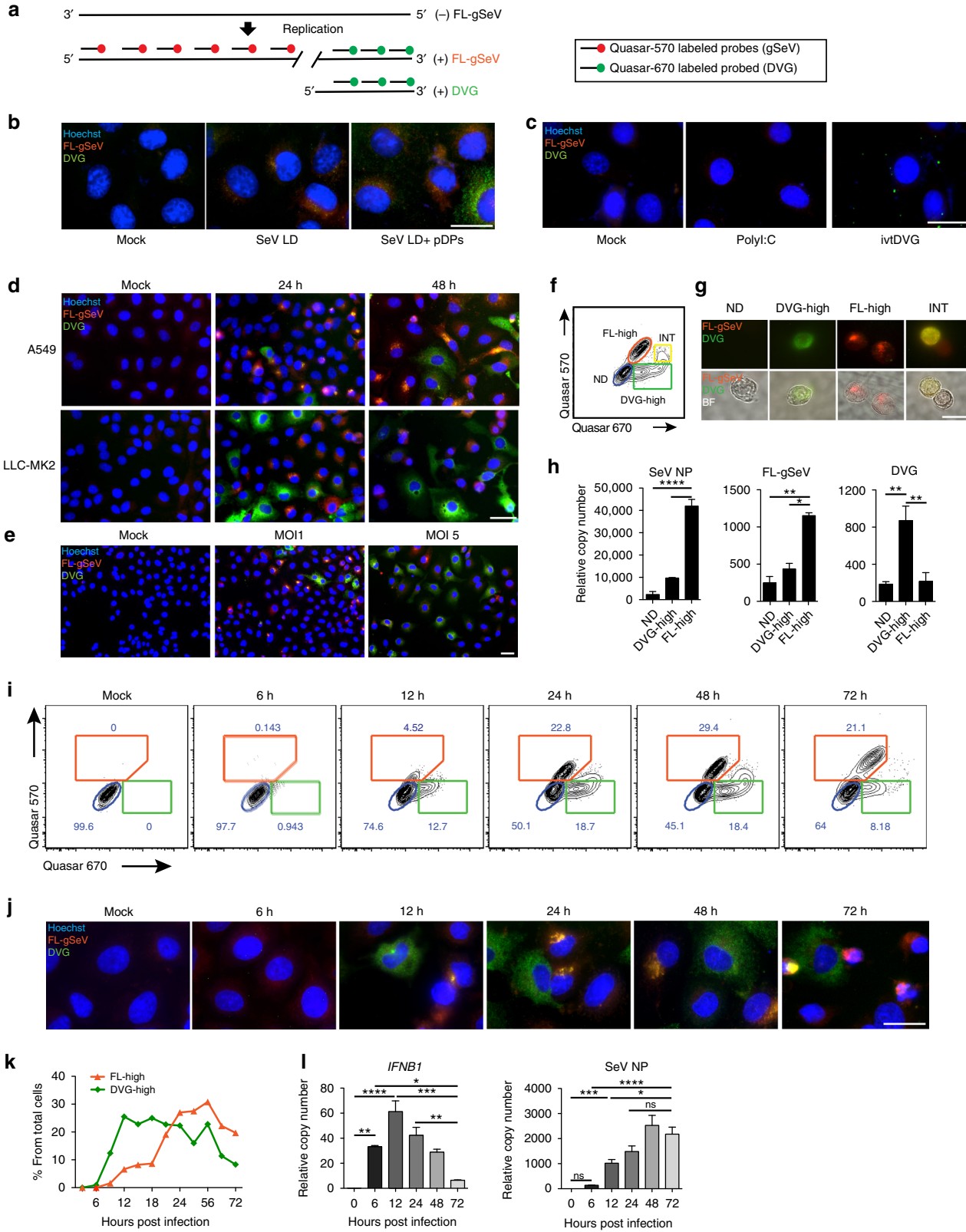

to 5.47% FL-high (Fig. 2c). Importantly, infectious virus was detected in the supernatant of survivor cells on day 17 post infection, confirming the generation of a persistent productive infection (Fig. 2d). In order to confirm a direct role for DVGs in promoting the generation of persistently infected cultures, pDPs were used to supplement SeV LD infections. As expected, extended survival of the infected population was rescued by supplementation with pDPs whereas supplementation with UV inactivated pDPs did not promote cell survival (Supplementary Fig. 2b–d). Similar to SeV HD-infected cells, cells that survived infection with SeV LD supplemented with pDPs contained replicative viral genomes and retained a significant population of DVG-high cells (Supplementary Fig. 2e, f). The predominance of DVG-high cell in the surviving cell population together with our inability to generate persistently infected LLC-MK2 cultures during infection with SeV LD virus stocks suggest that DVGs confer resistance to SeV-induced cell death. In support, the percentage of dead cells measured by viability staining was significantly higher in FL-high cells than DVG-high cells at 24 and 48 h post infection (Fig. 2e, f). Notably, the percentage of dead cells among the DVG-high sub-population did not significantly increase over time, whereas the percentage of dead cells among FL-high cells increased at 48 h post infection matching the spike of total cell death in the culture (Fig. 2g). To assess whether DVGs protect cells from apoptosis, a major mechanism of cell death during SeV infection[23, 24], we stained the cultures for the hallmark apoptotic molecules active caspase-3 and cleaved PARP-1. A significantly smaller population of DVG-high cells stained positive for these markers compared to FL-high cells (Fig. 2h, i, and Supplementary Fig. 2g) demonstrating that DVGs protect cells from apoptosis during infection.

To investigate whether the differential accumulation of DVGs and FL-genomes among infected cells and the associated phenotypes applied to other paramyxoviruses, we infected A549 cells with RSV having either high or low content of DVGs (HD and LD, respectively) and analyzed the cultures by RNA FISH using specific probes designed following the same strategy described for SeV. As with SeV HD infection, heterogeneous accumulation of RSV DVGs and full-length RSV genomes (FL-gRSV) was observed in the infected culture (Fig. 3a, b). Similar to SeV infection, cells infected with RSV LD did not generate distinguishable DVG signal (Fig. 3c). In addition, infection with RSV HD resulted in a persistently infected culture with survivor cells containing substantial amounts of both FL-gRSV and DVGs (Fig. 3c and Supplementary Fig. 3), while cells infected with RSV LD died after 1 week of infection (Fig. 3c, d). Further, DVG-high cells were protected from virus-induced cell death, as the percentage of apoptotic cells stained for active caspase-3 was significantly lower in DVG-high cells compared to FL-high cells (Fig. 3e, f). Taken together, DVG-

high cells are less prone to apoptosis than FL-high cells during SeV and RSV infection and extended cell survival associates with the establishment of persistent infections.

**DVG-high cells express a unique pro-survival transcriptome.** To characterize the mechanism protecting DVG-high cells from apoptosis, we identified candidate pathways by transcriptionally profiling sorted FISH-stained ND, FL-high, and DVG-high cells 24 h post SeV HD infection. A detailed description of the pipeline for these studies is presented in Supplementary Fig. 4a. Sorted populations with higher than 90% purity from multiple independent experiments were pooled after RNA extraction and used for total transcriptome profiling (Supplementary Fig. 4b, c). Confirming the distribution of viral genomes in the different populations, RNA-Seq analysis demonstrated even coverage of the full length genome in the FL-high population, while it showed a strong bias towards the 5′ end of the genome in the DVG-high population, as expected based on the origin of SeV copy-back DVGs (Supplementary Fig. 4d).

RNA-Seq analysis identified 1800 differentially expressed genes among the FL-high, DVG-high, and ND populations ($\geq$ two-fold change in expression with $\leq 1\%$ false discovery rate). Increased expression of a number of genes in the different cell populations was validated by RT-qPCR (Supplementary Fig. 4e). Hierarchical clustering revealed at least three distinct clusters of co-regulated genes. Clusters 1, 2, and 3 represent genes with relatively higher expression in ND cells, DVG-high cells, or FL-high cells, respectively (Fig. 4a). Gene Ontology analysis of cluster 1 (upregulated in ND cells) was highly enriched in genes involved in cell cycle checkpoints, complement and antigen presentation, and the IFN signaling pathway (Fig. 4b). Cluster 3 (upregulated in FL-high cells) showed enrichment in genes involved in the unfolded protein responses, oxidation–reduction pathways, and steroid metabolism (Fig. 4b). Cluster 2 (upregulated in DVG-high cells) demonstrated the enrichment of anti-apoptotic factors and a number of pro-survival pathways, including mitogens and growth factors, the TNF pathway, and the NF-kB/Rel pathway (Fig. 4b). Gene set enrichment analysis (GSEA) confirmed enriched signatures associated with pro-survival pathways in the DVG-high population (Fig. 4c, d). These signatures include TNF-related genes with anti-apoptotic functions, such as the TNF receptor 2 (*TNFR2* or *TNFRSF1B*), TNF receptor associated factor 1 (*TRAF1*), and the TNF alpha induced protein 3 (*TNFAIP3*, also known as A20)[25], molecules involved in regulation of NF-κB activity (for example, *NFKBIA* and *NFKBIE*)[26], and a number of genes involved in apoptosis regulation, including the baculoviral IAP repeat containing 3 (*BIRC3*, also know as *c-IAP2*)[27, 28] (Fig. 4d). In addition, the DVG-high population was enriched in genes involved in the

**Fig. 1** Differential accumulation of SeV full-length (*FL*) genomes and DVGs among infected cells. **a** RNA FISH strategy. Quasar-570-labeled probes (pseudocolored red) target the 5′ end of the SeV complementary genome (FL-gSeV), and Quasar-670-labeled probes (pseudocolored green) target the 3′ end of the SeV complementary genome and DVGs. Polarity of the genome is indicated. **b, c** RNA FISH images of A549 either **b** mock infected or infected with SeV LD alone or in the presence of pDPs (HAU = 300) for 8 h, or **c** transfected with in vitro transcribed (*ivt*) DVG or polyI:C. **d** RNA FISH images of A549 and LLC-MK2 cells infected with SeV HD. **e** A549 cells were infected with SeV HD at either a MOI of 1 or 5 TCID$_{50}$ per cell for 24 h. Representative RNA FISH images. **f** Representative RNA FISH-FLOW plot of A549 cells infected for 24 h with SeV HD. Gates correspond to non-detected (ND, *blue*), FL-gSeV-high cell (FL-high, *orange*), intermediate (INT, *yellow*), and DVG-high cells (DVG-high, *green*). **g** Representative images of sorted cells examined under the fluorescent microscope. Top: merged fluorescent images. Bottom: fluorescent images merged with bright fields (BF). **h** Expression of SeV NP mRNA, FL-gSeVs, and DVGs in sorted populations (*n* = 3). **i** Representative RNA FISH-FLOW plots of A549 cells at the indicated times after SeV HD infection. Gates correspond to FL-high cells (*orange*), ND (*blue*), and DVG-high cells (*green*). **j** RNA FISH images corresponding to cells analyzed in **i**. **k** Percentage of FL-high and DVG-high populations quantified from RNA FISH-FLOW shown in **i**. **l** Expression of cellular *IFNB1* mRNA and SeV NP mRNA in A549 during the infection time course. RT-qPCR data are expressed as the copy number relative to the housekeeping gene *GAPDH* mRNA. **h, l** Data are shown as mean ± s.e.m. *P < 0.05, **P < 0.01, ***P < 0.001, and ****P < 0.0001 by one-way ANOVA with Bonferroni's *post hoc* test. Images were taken at a ×20 (**e**), ×40 (**d**), or ×100 (**b, c, g, i**) magnification. Scale bar = 20 μm. See also Supplementary Fig. 1

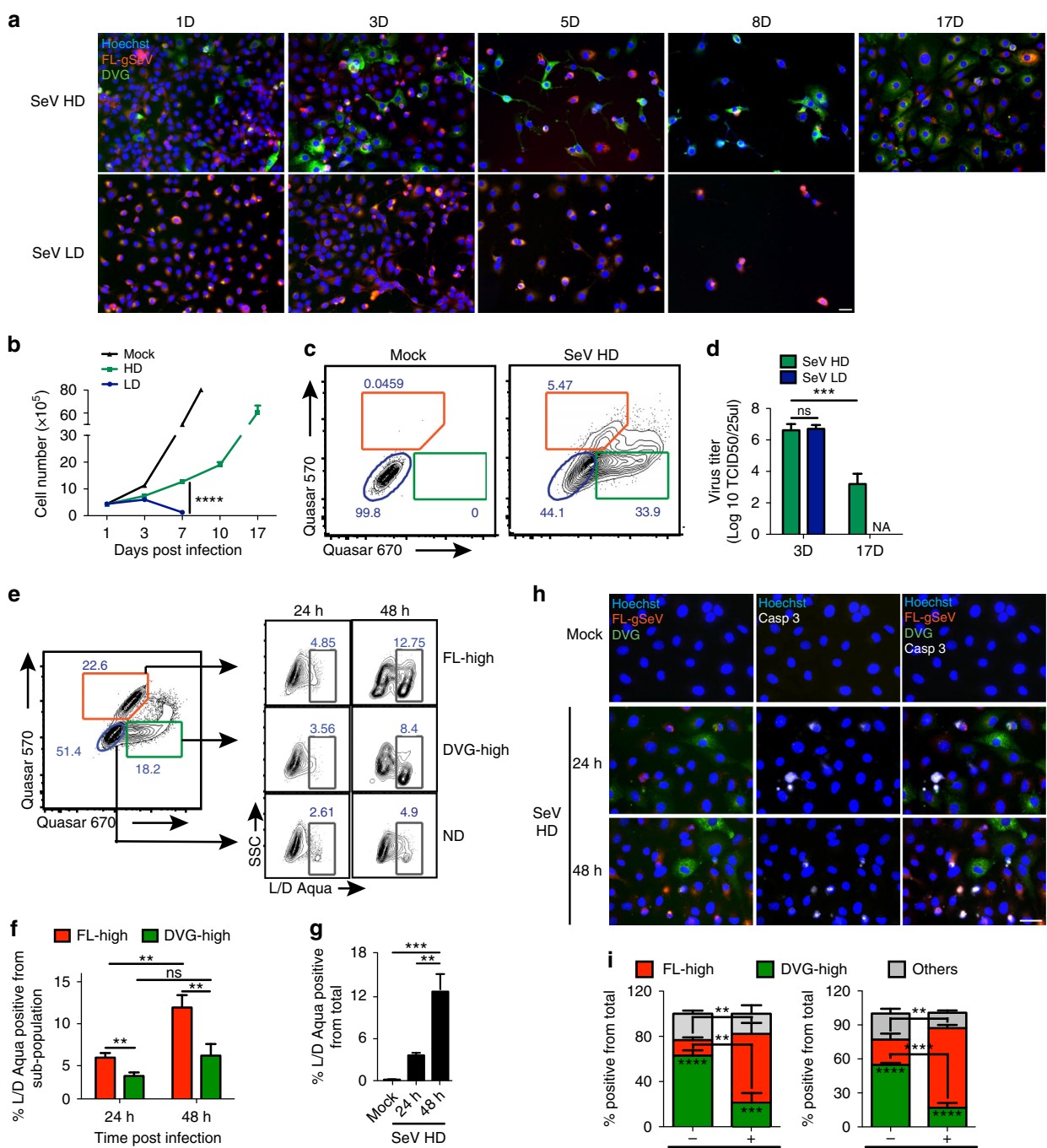

**Fig. 2** DVG-high cells show enhanced survival during SeV infection. **a** RNA FISH images of LLC-MK2 cells infected with SeV HD or SeV LD and sub-cultured for 17 days. **b** Total cell numbers during the infection time course ($n = 3$). **c** Representative RNA FISH-FLOW analysis of LLC-MK2 cells on day 17 after mock or SeV HD infection. The percentages of ND (blue), FL-high (orange), and DVG-high (green) in the culture are indicated. **d** Infectious virus titer in the supernatant of infected cells collected at the indicated time points ($n = 3$). **e** A549 cells were infected with SeV HD and stained at the indicated time points using the LIVE/DEAD viability assay (L/D Aqua) followed by RNA FISH-FLOW. Representative plots are shown. ND (blue), FL-high (orange), DVG-high (green), and dead cells (grey) gates are indicated. **f** Percentage of cells staining positive for L/D Aqua among each subpopulation ($n = 8$).
**g** Percentage of cells staining positive for L/D Aqua within the total population ($n = 8$). **h** Representative RNA FISH-IF image of active caspase-3 protein (white) staining of A549 cells infected with SeV HD for 24 or 48 h. Hoechst signals were merged with either RNA FISH probes signals (left column), active caspase-3 signal (Casp3, middle column), or all channels (right column). **i** Quantification of RNA FISH and IF staining for the apoptosis markers active caspase-3 (left panel) and cleaved PARP-1 (C-PARP-1, right panel) measured 24 h post infection. Graphs show the percentage of FL-high, DVG-high, and other cells that stained either positive or negative for the indicated apoptotic marker, $n = 3$–4. Stars inside the columns indicate significance between FL-high and DVG-high; stars between two columns indicate significance between active caspase-3 or C-PARP-1 negative and positive populations among FL-high or DVG-high sub-populations. Unless indicated, experiments were independently repeated at least three times. Data are shown as mean ± s.e.m. $*P < 0.05$, $**P < 0.01$, $***P < 0.001$ and $****P < 0.0001$ by **g** one-way or **b**, **d**, **f**, **i** two-way ANOVA with Bonferroni's post hoc test. ns, non-significant. NA, not available. Images were taken at a ×20 (**a**) or ×40 (**h**) magnification. Scale bar = 20 μm. See also Supplementary Fig. 2

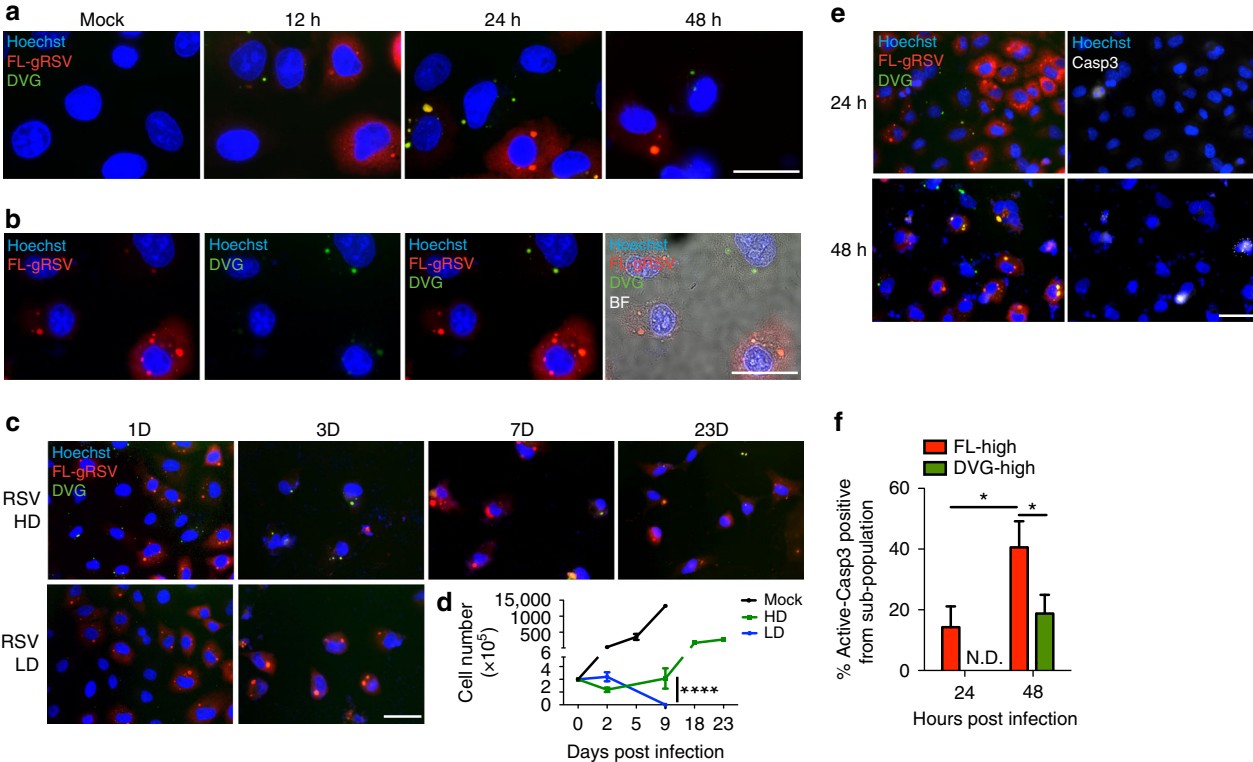

**Fig. 3** Heterogeneous accumulation of DVGs and FL-genomes during RSV infection. **a** Representative RNA FISH merged images of A549 cells infected with RSV HD for 12, 24, and 48 h. **b** Representative single channel and merged images of A549 cells infected with RSV HD for 24 h. Hoechst signal was merged with gRSV, DVG, or bright field (BF) respectively. **c** Representative RNA FISH merged images of A549 cells infected with RSV HD and RSV LD on days 1, 3, 7, and 23 post infection. **d** Total cell numbers of RSV HD and LD infected cells during the infection time course (n = 3). **e** Representative merged images of RNA FISH-IF for active caspase-3 protein (white) staining of A549 cells infected with RSV HD for 24 or 48 h. Hoechst signals were merged with either RNA FISH probes signals (left column) or active caspase-3 signal (Casp3, right column). **f** Percentage of active caspase-3 positive cells within FL-high and DVG-high cells infected as in **e** (n = 3). Data are shown as mean ± s.e.m. *P < 0.05, ***P < 0.001, and ****P < 0.0001 by two-way ANOVA with Bonferroni's post hoc test. ns, non-significant. ND, non-detectable. Images were taken at a × 40 (**c**, **e**) or ×100 (**a**, **b**) magnification. Scale bar = 20 μm. See also Supplementary Fig. 3

induction of IFNs and expression of cytokines and chemokines, consistent with a well-described role of DVGs in initiating antiviral immunity[9, 10, 22]. Collectively, these studies show that host cells enriched in DVGs actively engage both an antiviral and a pro-survival program.

**DVG-high cells survive apoptosis triggered by MAVS/TNF.** To determine if accumulation of DVG-high cells and death of FL-high cells depend on the antiviral response induced by DVGs[9, 10], we analyzed infections of cells lacking the critical adaptor protein MAVS (MAVS KO) or lacking the type I IFN receptor (IFNAR1 KO). Differential accumulation of DVG-high and FL-high cells remained unchanged in MAVS or IFNAR1 KO cells, suggesting that the heterogeneous accumulation of DVGs is independent of the engagement of the antiviral response (Fig. 5a). In addition, the percentage of apoptotic cells in both DVG-high and FL-high populations, remained similar in wild type (WT) and IFNAR1 KO cells (Fig. 5a–c). In contrast, MAVS KO cells exhibited reduced apoptosis in both subpopulations (Fig. 5a–c), indicating that a MAVS-dependent process drives apoptosis in these cultures. The lower level of apoptosis in SeV HD-infected MAVS KO cells compared to controls was not due to reduced viral replication, as MAVS KO cells showed increased viral replication (Supplementary Fig. 5a, b).

Transcriptional profiling suggested that TNFα-related pathways are implicated in defining the cell fate during SeV infection. Due to the well-documented dual roles of TNFα in both inducing

cell death and conferring survival[29], we next investigated whether TNFα signaling orchestrates the differential pro- and anti-apoptotic responses of FL-high and DVG-high cells. As expected, SeV infection induced high levels of TNFα protein and mRNA in a MAVS-dependent but IFNAR1-independent manner (Fig. 5d and Supplementary Fig. 5c). The absence of TNFα secretion in MAVS KO cells correlated with their decreased level of apoptosis in response to SeV infection (Fig. 5a, b), and supplementation of MAVS KO cells with TNFα, but not IFNβ, significantly increased apoptosis upon infection with SeV HD (Fig. 5e) or LD (Supplementary Fig. 5d, e). Moreover, a combination of neutralizing antibodies against TNFα and its receptors significantly reduced the level of total active caspase-3 expression (Fig. 5f and Supplementary Fig. 5f), indicating that the MAVS/TNF axis regulates cell death in these cultures.

To investigate whether TNFα differentially impacts apoptosis of FL-high and DVG-high cells, we quantified the percentage of active caspase-3 positive cells among each sub-population during infection in the presence of neutralizing antibodies. Unexpectedly, while it increased apoptosis of DVG-high cells, blocking TNFα signaling significantly reduced apoptosis of FL-high cells, indicating that TNFα signaling protects DVG-high cells and kills FL-high cells (Fig. 5g). Interestingly, supplementation of MAVS KO cells with increasing doses of TNFα enhanced apoptosis in the infected culture (Fig. 5h), but did not phenocopy the differential effect on cell death of FL-high and DVG-high cells seen in WT infected cells as both sub-populations were susceptible to TNFα-induced apoptosis to a similar extent at all

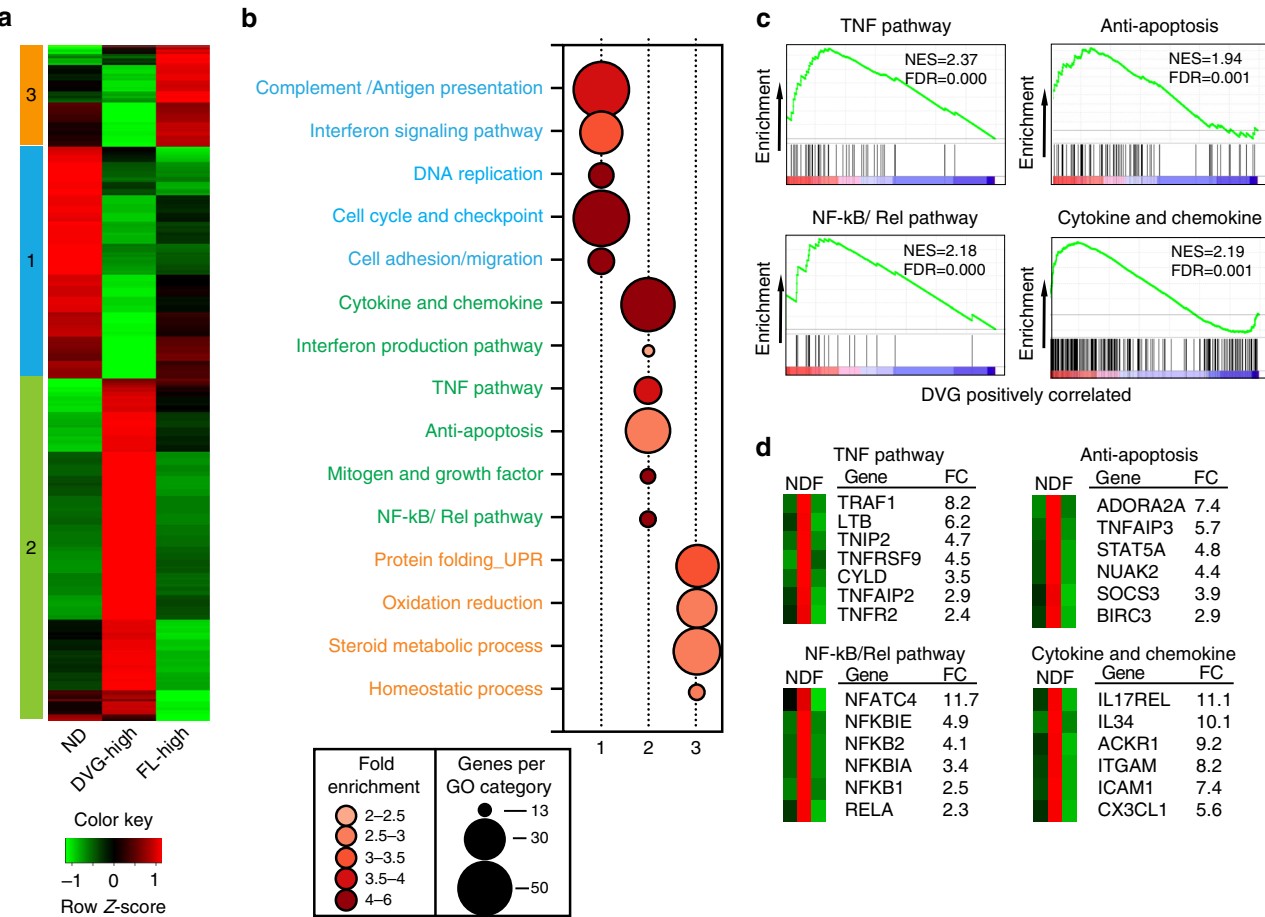

**Fig. 4** Transcriptional profiling of cells sorted based on DVG and FL-gSeV abundance identified distinct pro-survival pathways in DVG-high cells. **a** Hierarchical clustering and heatmap representation of 1800 genes differentially regulated among ND, FL-high, and DVG-high populations sorted from SeV HD-infected A549 cells at 24 h post infection. Color based on row Z-score. Three clusters of distinctly upregulated genes in each of the sorted sub-populations are indicated. **b** Bubble chart showing Gene Ontology (GO) enrichment analysis. Bubble size indicates the number of genes associated with each term. Bubble color intensity indicates fold enrichment of GO terms overrepresented in that cluster of genes. Pathways upregulated in each cluster were color-coded as blue (Cluster 1), green (Cluster 2) and orange (Cluster 3). **c** GSEA enrichment plots for four selected pathways differentially regulated in the DVG-high cells compared with others. Normalized enrichment scores (NES) and false discovery rate (FDR) are indicated. **d** Selected genes from each GSEA signature panel shown in **c**. Average fold change (FC) between DVG-high and FL-high cells is shown. N = ND, D = DVG-high, and F = FL-high in the mini-heatmaps shown in the left side of each graph. See also Supplementary Fig. 4

doses tested (Fig. 5h, i). Taken together, these results suggest that DVG-high cells lose protection from TNFα-mediated apoptosis in absence of MAVS and suggest a direct role of MAVS signaling in promoting the survival of DVG-high cells.

**MAVS signaling dictates the survival of DVG-high cells**. We next investigated how DVG-high cells become less prone to TNFα-mediated apoptosis compared to FL-high cells within the same infected culture. Based on RNA-Seq data showing upregulation of several genes in the TNFR2 signaling pathway in DVG-high cells, including TNFR2 itself (*TNFRSF1B*) and *TRAF1*, we hypothesized that the TNFR2 pathway is responsible for the protection of DVG-high cells from virus-triggered apoptosis. To test this hypothesis, we neutralized either the primary TNF receptor TNFR1 or TNFR2 in SeV HD-infected cultures (Fig. 6a, b) and tested for apoptosis of the different sub-populations. Treatment with TNFR2 neutralizing antibodies significantly increased apoptosis of DVG-high but not FL-high cells, suggesting a pro-survival function of TNFR2 signaling in cells enriched in DVGs. In contrast, blocking TNFR1 signaling reduced cell death in FL-high but had no impact on the survival

of the DVG-high population (Fig. 6a, b) suggesting that TNFR1 signaling is not necessary for DVG-survival. In agreement with a pro-survival role for TNFR2 in DVG-high cells, knock-down of the TNFR2 adaptor molecule TRAF1 or the downstream effector BIRC3 increased apoptosis in DVG-high cells but not in FL-high cells (Fig. 6c, d and Supplementary Fig. 6). Interestingly, DVG-mediated upregulation of TNFR2 signaling was controlled by MAVS, as MAVS KO cells showed impaired upregulation of surface TNFR2 expression and blunted expression of the pro-survival genes *TRAF1*, *BIRC3*, and *TNFAIP3* upon infection with SeV HD (Fig. 6e, f). Thus, the TNFR2 pathway, which is upregulated in DVG-high cells in a MAVS-dependent manner, protects this population from SeV-induced TNFα-mediated apoptosis. Furthermore, treatment of SeV HD-infected cells with TNFR2 neutralizing antibody led to massive cell death, similar to infection with SeV LD, but cells that eventually recovered showed a significantly reduced percentage of infection after long-term subculture compared to untreated cells (Fig. 6g–j). Together, these data demonstrate that engagement of the MAVS/TNFR2 axis by DVGs allows a sub-population of infected cells to survive TNFα-mediated apoptosis, which is exploited by the virus to achieve long-term persistence.

## Discussion

In this study we demonstrate that while cells respond to DVGs inducing the expression of antiviral molecules that restrict virus production, they also engage a pro-survival TNFR2/TRAF1-dependent mechanism that promotes viral persistence. Our data support a model in which enrichment of DVGs over full-length viral genomes in infected cells leads to engagement of a MAVS-dependent pathway that while driving antiviral and pro-apoptotic activities through the production of IFNs and TNFα, protects the cytokine-producing cell from death by inducing and engaging TNF-related pro-survival factors (Fig. 6k). These observations not only explain a long-standing paradox between the reported immunostimulatory and pro-persistence activities of DVGs, but also reveal complex host-pathogen

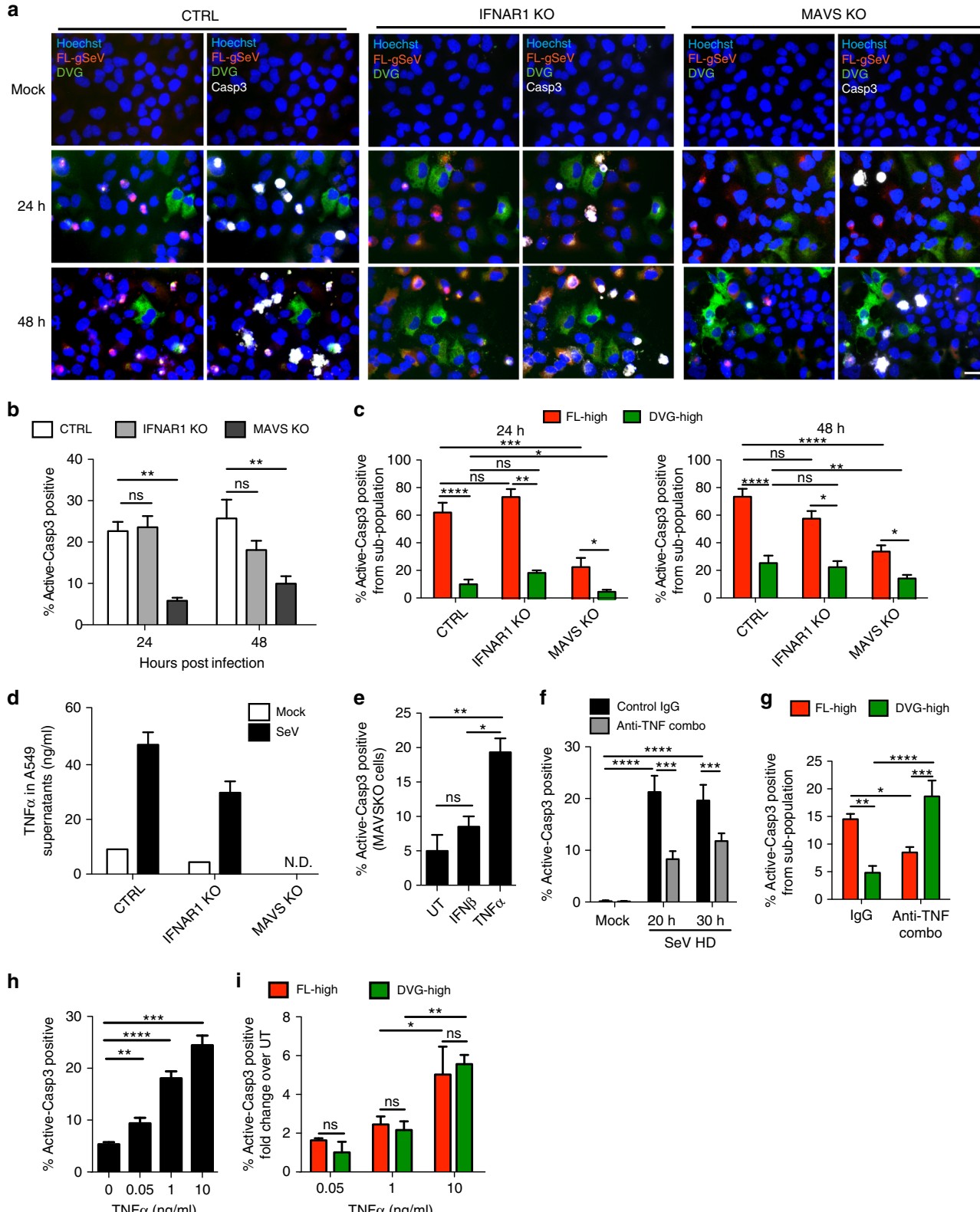

interactions that may, in part, explain the co-existence of viruses and their hosts in immunocompetent individuals.

The testing and validation of this model was only possible after developing the ability to identify and distinguish DVGs from FL-genomes at a single cell level. Prior to our imaging approach, DVGs were usually detected by PCR or northern blot in bulk infected cultures, masking the heterogeneity of viral genome distribution among infected cells, as well as the distinct cellular responses towards the infection. Of note, given the diversity of DVGs, it is likely that distinct DVG species are generated during infection and that the ability of different DVGs to promote virus persistence varies. Our study assessed exclusively the function of copy-back DVGs. As copy-back DVGs confer the most potent immunostimulatory activity to paramyxovirus infections we considered it relevant to follow their activity, regardless of the potential function of other DVGs. Moreover, we provide evidence that copy-back DVG-546 present in cultures infected with SeV HD[8, 22, 30] is the predominant copy-back species detected in our assays (Supplementary Fig. 4d). RSV, in contrast, produces a discrete population of copy-back DVGs[5] and our probes are designed to capture the large majority of these copy-back DVGs. Interestingly, in A549 and LLC-MK2 cells the DVG signal was widely spread within the cell cytoplasm, while the FL-gSeV signal was concentrated in a perinuclear region of the infected cells (Fig. 1d, j) indicating a differential intracellular localization for FL-gSeV and DVGs. In agreement, in cells infected with SeV lacking DVGs (SeV LD), FL-gSeVs were concentrated in the perinuclear region (Fig. 2a and Supplementary Fig. 1a). The impact of the intracellular distribution of viral genomes is unknown and is currently under investigation.

DVG specific RNA FISH revealed two phenomena that challenge the prevailing view of the temporal and spatial dynamics of DVGs during infection and provide new insights into their contribution to virus persistence. First, instead of a homogenous distribution of DVGs together with FL-viral genomes among infected cells, DVGs predominantly occupied a sub-population of the infected cells. Second, changes in the percentage of DVG-high cells throughout the course of infection coordinated well with changes in the percentage of FL-high cells, providing an alternative explanation for the long standing observation of waves of DVGs and FL-viral genome predominance in infected cultures. Our evidence of distinct cellular populations based on viral genomic content and the resulting differential susceptibility to cell death reveal an underappreciated cellular level of regulation of persistent viral infections.

Our data demonstrate that during virus infection cells with a high content of FL-gSeV are susceptible to TNFα but not type-I IFN mediated apoptosis. The apparent lack of pro-apoptotic activity of IFN over FL-high cells is likely caused by the strong IFN signaling antagonistic activity of paramyxovirus proteins[31]. TNFα is a pro-inflammatory cytokine that can be induced by viral

infection and plays important roles in the control of virus dissemination[32, 33]. Expression of TNFα has been reported in many paramyxovirus infections, including RSV, SeV, Newcastle disease viruses, and parainfluenza virus 5 (also known as simian virus 5)[34, 35]. Importantly, TNFα expression is stimulated by DVGs in both SeV and RSV infections[5, 22]. In addition, purified defective viral particles from SeV can cause selective apoptosis of transformed cells by activating the TNF-related apoptosis-inducing ligand (TRAIL) both in vitro and in vivo[36]. Together, this evidence indicates that DVGs can drive TNF-mediated apoptosis of virus-infected cells through a mechanism distinct from that leading to cell death after LD infections, likely caused by virus-induced cell stress.

Unexpectedly, transcriptome analysis revealed a robust pro-survival TNF signature in DVG-high cells. In addition, neutralization of TNFα and its receptors during SeV infection resulted in significantly increased apoptosis of DVG-high cells confirming a critical requirement for cell extrinsic TNF signaling in promoting their survival. The enhanced survival of DVG-high cells compared to FL-high cells could potentially also be explained by reduced lytic virus replication in DVG-high cells as DVGs interfere with the replication of the FL-gSeV by competing for the usage of the viral polymerase[16, 17]. However, both SeV- and RSV-induced apoptosis were significantly reduced in MAVS KO cells despite similar or even higher levels of replication compared to WT controls, indicating that virus replication is not the sole determinant of cell fate during infection. The pro-survival response to TNFα signaling is a well-documented protective mechanism mediated by NF-kB induced upregulation of pro-survival molecules, including TRAF1, A20 (TNFAIP3), and cIAP2 (BIRC3)[37–40]. Here we identified a TNFR2/TRAF1 mediated pathway as essential to the survival of DVG-high cells. Notably, the expression and activity of key elements in this pathway, including TNFR2 and TRAF1, required the engagement of MAVS signaling in DVG-high cells and in the absence of MAVS TNFα treatment rendered DVG-high and FL-high cells equally susceptible to apoptosis. This discovery uncouples the role of the TNF pathway in dictating the individual cell fate during infection and identifies the interaction of DVGs with the MAVS-mediated antiviral pathway as a critical factor in defining cell survival upon infection.

The mechanism driving the heterogeneity of FL genomes and DVG distribution and the associated innate immune responses during initial viral infection remains unclear. Cell to cell variation in type I IFN expression has been demonstrated in a number of systems in association with differential expression of innate immune limiting factors[41, 42]. These studies imply that select host factors may control differential responses towards virus infection. Interestingly, our data show that the heterogeneous distribution of FL genomes and DVGs among infected cells along with the associated differential cell death rate is not

**Fig. 5** Distinct TNF pathways determine the cell's fate during virus infection. **a–c** A549 CRISPR control (CTRL), IFNAR1 KO, and MAVS KO cells mock-infected or infected with SeV HD for 24 or 48 h. **a** Merged representative RNA FISH-IF images of active caspase-3 positive staining. Magnification: ×40. Scale bar = 20 μm. **b** Percentage of active caspase-3 positive cells within each infected culture (n = 8). **c** Percentage of active caspase-3 positive cells within the FL-high and DVG-high populations of each culture (n = 8). **d** TNFα in the supernatant of A549 CTRL, IFNAR1 KO, and MAVS KO cells mock infected or infected with SeV for 24 h. One representative experiment is shown, bar = mean ± s.e.m. of technical triplicate. **e** Percentage of active-caspase 3 positive MAVS KO cells infected with SeV HD for 16 h and either left un-treated (UT), treated with recombinant IFN-beta (IFNβ, 100U), or treated with recombinant TNFα (10 ng/ml) for 8 h. (n = 3). **f, g** A549 cells were infected with SeV HD and incubated with control IgG antibody or antibodies against TNFα, TNFR1, and TNFR2 (anti-TNF Combo). **f** Percentage of apoptotic A549 cells from the total population (n = 4). **g** Percentage of active caspase-3 positive cells within FL-high and DVG-high cells (n = 4). **h, i** MAVSKO A549 cells were infected with SeV HD for 16 h and then incubated with recombinant TNFα at 0, 0.05, 1, and 10 ng/ml for 8 h. **h** Percentage of active caspase-3 positive cells among total cells (n = 3–6). **i** Percentage of active caspase-3 positive staining cells within FL-high and DVG-high cells (n = 3). Data are shown as mean ± s.e.m. of fold change over UT at each TNFα dosage. *P < 0.05, **P < 0.01, ***P < 0.001, and ****P < 0.0001 by one-way (**e, h**) or two-way (**b, c, f, g, i**) ANOVA with Bonferroni's post hoc test. ns, non-significant. ND, non-detected. See also Supplementary Fig. 5

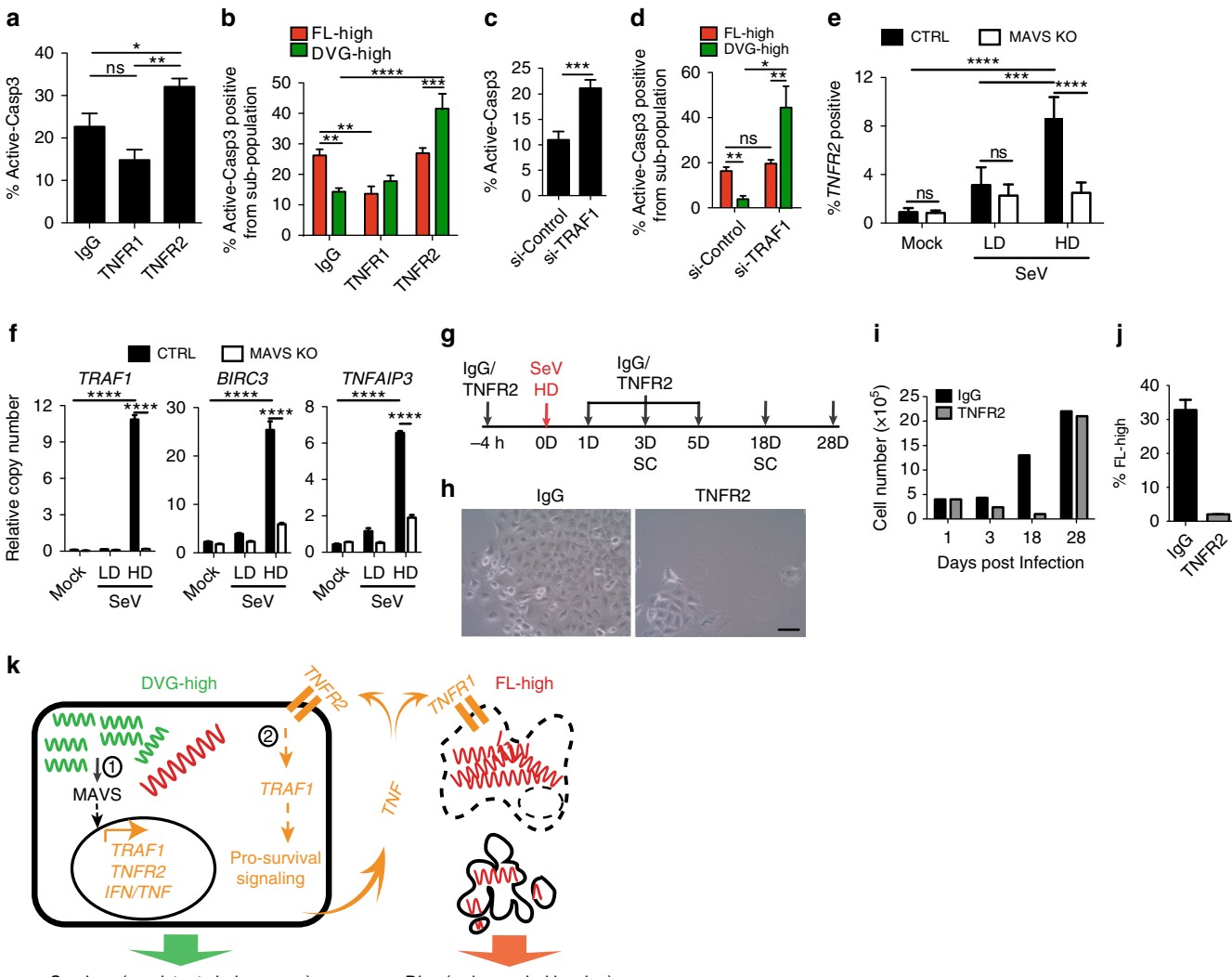

**Fig. 6** The survival response of DVG-high cells is regulated by MAVS signaling. **a**, **b** SeV HD-infected A549 cells treated with either control-IgG, TNFR1 or TNFR2 neutralizing antibodies. **a** Percentage of active caspase-3 positive cells within the total cultures ($n = 4$). **b** Percentage of active caspase-3 positive cells within FL-high and DVG-high cells in each condition ($n = 4$). **c**, **d** A549 cells transfected with either control siRNA (si-Control) or *TRAF1* siRNA (si-*TRAF1*) followed by SeV HD infection for 24 h. **c** Percentage of apoptotic cells within the total cell cultures ($n = 3$). **d** Percentage of active caspase-3 positive cells within FL-high and DVG-high cells in each condition ($n = 3$). **e**, **f** A549 CTRL and MAVS KO cells infected with SeV LD and HD (MOI = 1.5 TCID$_{50}$ per cell) for 24 h. **e** Expression of surface TNFR2 was quantified by flow cytometry ($n = 4$) and **f** expression of cellular *TRAF1*, *BIRC3*, and *TNFAIP3* (A20) mRNA by RT-qPCR. Data are expressed as copy number relative to the housekeeping gene *GAPDH* ($n = 4$). **g**–**j** A549 cells treated with control IgG (IgG) or TNFR2 neutralizing antibodies (TNFR2) during SeV HD infection were sub-cultured (SC) for 28 days. **g** Illustration of the experiment schedule. **h** Bright field images 18 days post SeV HD infection. One representative field from each treatment is shown. Magnification: ×20. Scale bar = 20 μm. **i** Total cell numbers during the infection time course. **j** Percentage of FL-high cells quantified by RNA FISH-FLOW 28 days post infection. Experiments were repeated twice and one representative experiment is shown. **k** Schematic representation of proposed model. DVGs-high cells engage a MAVS-dependent antiviral pathway that drives the antiviral response and the apoptosis of neighboring cells through the production of TNFα (denoted 1), while protecting the cytokine-producing cell from death through a TNFR2/TRAF1 mediated mechanism (denoted 2). Data are shown as ± s.e.m. *$P < 0.05$, **$P < 0.01$, ***$P < 0.001$, and ****$P < 0.0001$ by one-way (**a**) or two-way (**b**, **d**, **e**, **f**) ANOVA with Bonferroni's post hoc test, or unpaired *t*-test **c**. ns, non-significant. See also Supplementary Fig. 6

significantly altered in IFNAR1 KO or MAVS KO cells (Fig. 5a), suggesting that factors additional to the type I IFN pathway are involved. One possibility is that the cell cycle status at the time of infection is a critical factor in determining heterogeneity. Another possibility is that stochastic DVG accumulation in infected cells drives the phenotype. In this case, it is expected that DVG accumulation is a dominant phenotype and when cells are seeded with DVGs, either through stochastic infection with viral pDPs or through generation of DVGs during virus replication, DVGs take over the viral replication process. Interestingly, our preliminary experiments show that natural accumulation of DVGs during SeV

replication in vivo follows a similar heterogenic pattern, where some cells in the lung epithelium accumulate DVGs and others do not, demonstrating that this process is not limited to in vitro infections.

Overall, here we describe an intricate mechanism in which virus and host come to a balance to establish a symbiotic interaction. The virus benefits from this mechanism because it allows extended survival of virus-infected cells, establishing a persistent infection. This relationship is founded in the dual role of TNFα in targeting highly infected cells (FL-high) for apoptosis while protecting cells with active MAVS signaling (DVG-high) from

cell death. This mechanism fundamentally changes our understanding of the relationship between antiviral responses and viral persistence and reveals potential therapeutic targets to eliminate persistent viral reservoirs.

## Methods

**Cell cultures and stable cell lines**. A549 cells (human type II alveolar cells, ATCC, #CRM-CCL185), LLC-MK2 (monkey kidney epithelial cells, ATCC, #CCL-7), and control, MAVS KO, and IFNAR1 KO A549s were cultured at 5% $CO_2$ and 37 °C conditions with Dulbecco's modified Eagle's medium supplemented with 10% fetal bovine serum, 1 mM sodium pyruvate, 2 mM L-Glutamine, and 50 mg/ml gentamicin. KO A549 cells were generated using CRISPR/Cas9 as follows: The oligonucleotide sequences sgMAVS-for: CACCGGAGGGCTGCCAGGTCAG AGG and sgMAVS-rev**:**AAACCCTCTGACCTGGCAGCCCTC (IDT); sgIFNAR1 (set 1) for: CACCGGACCCTAGTTGCTCGTCGCCG and sgIFNAR1 rev: AAACC GGCGACGAGCAACTAGGGTC; sgIFNAR1 (set 2) for: CACCGTGGGTGTTGT CCGCAGCCGC and sgIFNAR1 rev: AAACGCGGCTGCGGACAACACCCA (Invitrogen) were used to generate stable guide RNAs (sgRNA) to target MAVS or IFNAR1 genes. The forward and reverse oligonucleotides were inserted into the plasmid vector pLenti-CRISPR (Addgene, Plasmid #52961) using BsmBI restriction sites. The resulting plasmids were packaged in pseudo lentiviral particles and used to establish the gene KO as previously described[43]. KO of MAVS was confirmed by western blotting (Santa Cruz, sc-166538) and mutations in the gene were verified by PCR using the following primers: MAVS-For: CTCCCCTGGCTCCTGTGCT CC and MAVS-Rev: AACTCCCTTTATTCCCACCTTG. Specific elimination of MAVS activity was confirmed upon transfection of 0.25 µg/$10^5$ cells of MAVS-expression plasmid (MAVS-WT, Addgene, Plasmid #52135) into KO cells and measuring antiviral gene expression 16 h post infection (Supplementary Fig. 5a). In addition, most experiments were repeated in a second independently generated MAVS KO cell line. Specific elimination of IFNAR1 was confirmed by flow cytometry (anti-hIFNAR1 antibody kindly provided by Serge Y. Fuchs, University of Pennsylvania) after IFNα treatment of cells and mutations were verified by PCR using the primers IFNAR1-For: GCTAGCTAGGAGGAAAGGCG and IFNAR1-Rev: GGGTTTAATCTTTGGCGCGG. All cell lines used in this study were authenticated by Short Tandem Repeat (STR) Profiling at ATCC, treated with mycoplasma removal agent (MP Biomedicals), and monthly tested for mycoplasma before use.

**Viruses**. Sendai virus strain Cantell HD and LD stocks (SeV HD, high DVG particle content; SeV LD, low DVG particle content) were prepared in embryonated chicken eggs as described previously[9, 10]. SeV LD used in this study had a medium tissue culture infectious dose ($TCID_{50}$)/HA titer of 98120 and SeV HD a $TCID_{50}$/HA titer of 4916. RSV LD (stock of RSV derived from strain A2, ATCC, #VR-1540 with a low content of DVGs) and RSV HD (derived from RSV LD) were prepared and characterized as described previously[44]. For SeV titration, LLC-MK2 cells (ATCC, #CCL7) were infected with triplicate serial 1:10 dilutions of the virus stock or supernatant from infected cell cultures (Fig. 2d only) in the presence of 2 mg/ml of trypsin. After 72 h of incubation at 37 °C, supernatants from each well were tested by hemagglutination of chicken RBCs for the presence of virus particles at the end point dilution as describe in ref. [9]. For virus infections in vitro, cells were incubated with virus at a multiplicity of infection (MOI) of 1.5 $TCID_{50}$ per cell unless otherwise indicated in the figure. Of note, to maintain consistency, the MOI was calculated throughout the manuscript based on the titration of virus stocks in LLC-MK2 cells, regardless of the cell type infected.

**Purified defective particles preparation**. Purified defective particles (pDPs) preparation was performed as previously described[10]. Briefly, allantoic fluid from 100 SeV Cantell-infected embryonated eggs was concentrated by high-speed centrifugation. The resulting pellets were suspended in 0.5 ml of PBS/2 mM EDTA and incubated overnight at 4 °C. The suspension was then added on top of a 5–45% endotoxin free sucrose (Fisher) gradient prepared using a gradient maker (BioComp). Gradients were centrifuged at 28,000 rpm (133,907 × g) for 1.5 h and the fraction containing low-density viral particles was collected and re-purified using the same procedure. Final low-density fractions were concentrated by centrifugation at 21,000 r.p.m. (75,619 × g) for 2 h. Pellets were suspended in PBS and stored at −80 °C. The content of pDPs particles was determined by calculating the $TCID_{50}$/HA ratio as reflection of infectious over non-infectious particles as described previously[10]. pDPs used in this study had a $TCID_{50}$/HA ratio of 96.9.

**RNA extraction and RT-qPCR**. Total RNA was extracted using TRIzol (Invitrogen) according to the manufacturer's specifications and 1–2 µg of RNA was reverse transcribed using the High Capacity RNA-to cDNA kit (Applied Biosystems). cDNA was amplified with specific primers in the presence of SYBR green (Applied Biosystem). qPCR reactions were performed in triplicates using an Applied Biosystem ViiA7 Real-time Lightcycler. For the primers used, see Supplementary Table 1. For FL-gSeV and SeV DVGs detection, 1–2 µg of isolated total RNA were reverse transcribed with specific primers for DVG (5′ GGTGAGGAATCTATAC GTTATAC 3′) and FL-gSeV (5′-TGTTCTTACTAGGACAAG-3′) using

Superscript III without RNaseH activity to avoid self-priming. Recombinant RNase H (Invitrogen) was later added to the reverse transcribed samples. cDNA was amplified with viral product-specific qPCR primers listed in Supplementary Table 1 in the presence of SYBR green (Applied Biosystem). Copy numbers were normalized to the housekeeping gene *GAPDH*. RSV DVGs were detected by PCR as described in ref. [5].

**Preparation of in vitro transcribed DVG RNA**. To generate in vitro transcribed SeV DVG RNA, a plasmid expressing SeV-DVG (pSL1180-DVG546 described previously[45], was linearized and in vitro transcribed using the MEGAscript T7 kit (Ambion) in the presence of RNase inhibitors (Invitrogen). The resulting products were then treated with DNase and then precipitated with LiCl (both included in the MEGAscript T7 kit). All in vitro transcribed DVG RNA had a optical density 260 nm ($OD_{260}$)/$OD_{280}$ ratio of 2.00–2.25, and a $OD_{260}$/$OD_{230}$ ratio of 2.20–2.60.

**RNA FISH and immunofluorescence staining**. RNA fluorescence in situ hybridization (RNA FISH) was performed according to published protocols with some modifications[45]. The probes used for both SeV and RSV detection were single stranded DNA oligos (20 nucleotides) each labeled with one fluorophore (Quasar 570 or Quasar 670, Biosearch Technologies). Briefly, probes detecting positive sense (+) FL-gSeV genome and most viral mRNA targeted position 1 to 11,151 of the SeV genome (GenBank: AB855654.1, excluding the 5′ end that encompasses the DVG sequences) and labeled with Quasar 570; probes detecting positive sense (+) DVGs targeted position 14,944 to 15340 of the SeV genome and were labeled with Quasar 670 (targeted sequence for each probe is listed in Supplementary Table 2). For detecting negative sense FL-gSeV, a pool of Quasar-570-labeled probes target position 1 to 13,685 of the negative sense FL-gSeV was used (targeted sequence for each probe is listed in Supplementary Table 3). For RSV FISH probes: a pool of 32 probes detecting positive sense FL-gRSV genome were designed targeting from position 1 to 11959 of the RSV genome (GenBank: AF035006, excluding the 5′ end that encompasses the RSV DVG sequences) and labeled with Quasar 570; probes detecting positive sense copy-back RSV DVGs targeted position 14,923 to 15,222 of the RSV genome and were labeled with Quasar 670 (targeted sequence for each probe is listed in Supplementary Table 4).

For RNA FISH, cells were plated onto coverslips (Corning) at a density of $4 \times 10^5$ cells per well in six-well plates and grown overnight at 37 °C. The cells were washed once with ice-cold PBS followed by fixation with 4% formaldehyde in PBS for 10 min and then permeabilized in 70% ethanol. Fixed cells were then equilibrated in wash buffer containing 10 % formamide and 2× saline sodium citrate (SSC, 1× SSC is 0.15 M NaCl plus 0.015 M sodium citrate, all from Thermo Fisher Scientific). RNA FISH was performed by hybridizing fixed cells with probes (125 nM for SeV and 250 nM for RSV) diluted in 50 µl hybridization buffer consisting of 10 % formamide, 2× SSC, and 10 % (wt/vol) dextran sulfate. Hybridization was performed overnight in a humidified chamber at 37 °C. Nuclear staining using 0.5 µg/ml of Hoechst 33342 (Invitrogen) was performed afterwards and the coverslides were mounted in GLOX anti-fade media (10% glucose, 1 M Tris-HCl pH8.0, glucose oxidase, catalase, diluted in 2× SSC; all from Sigma) before imaging. For RNA FISH coupled with immunofluorescence (IF) (RNA FISH-IF), fixed cells were permeabilized with 70% ethanol for 1 h. Permeabilized cells were then incubated with anti-human active caspase-3 antibody (1:100 dilution; Cell Signaling, cat. no. 9661) or anti-human cleaved PARP-1 antibody (1:100 dilution; Cell Signaling, cat. no. 5625) followed by Alexa Fluor 488-labeled goat anti-rabbit IgG (1:500 dilution; Invitrogen, cat. no. R37116) diluted in 1% BSA in the presence of 40 U/ml RNase inhibitor (Invitrogen, cat. no. 10777-019). Stained cells were incubated in 4% formaldehyde for 10 min prior to RNA FISH, washed with PBS, and then equilibrated in wash buffer. RNA FISH was then performed as described above.

**Wide-field and fluorescence microscopy and image processing**. Fluorescence imaging acquisition was performed with a Nikon E600 epifluorescence microscope equipped with a × 20, × 40 and a × 100-1.4 numerical aperture oil immersion objective (Zeiss) and a Zeiss AxioCam MRm camera. Bright field image acquisition was performed on an Olympus CKX41 inverted microscope equipped with a ×20 objective and a Spot Idea 5 MP Scientific Digital camera system (Diagnostic Instrument). Images were processed using Volocity LE (PerkinElmer).

**Image analysis and quantification**. Image analysis and quantification were performed using the Meta-morph software (Molecular Devices). Exposure time, gain, and offset were held constant for all images. The multi-wavelength cell-scoring module was applied to determine average fluorescent intensity of Quasar 570, Quasar 670 probe signals, or targeted host protein signals in each cell. Pixel brightness of the signals in multiple empty areas of each image was used as a reference to set the threshold for positive staining of probes/antibodies. Because of the bi-color FISH probe design strategy for distinguishing DVG from FL-gSeV, each individual cell was quantified for their viral RNA content as a ratio of Quasar 670 versus Quasar 570 probe signal fluorescent intensity (gSeV/DVG or gRSV/DVG). For image quantification of SeV infected cells, a ratio below 0.8 was considered as FL-high cells, between 0.8 and 1.0 as intermediate (INT) and above 1.0 as DVG-high. For image quantification of RSV infected cells, a ratio

below 0.5 was considered as FL-high cells, between 0.5 and 0.8 as INT and above 0.8 as DVG-high. For quantifying FISH coupled with active caspase-3 staining images, same threshold value of gSeV/DVG or gRSV/DVG ratio was used to categorize cells into FL-high or DVG-high. Positive scoring using a set threshold for positive staining identified active-caspase 3 positive and negative cells. The percentage of active caspase-3 positive cells among FL-high and DVG-high cells was then calculated based on the score. All image quantifications shown were obtained from at least 500 cells from four different fields in each repeat through at least three independently repeated experiments.

**RNA FISH-FLOW and FACS sorting**. The procedure described previously for RNA-FISH of cells fixed in slides was used for RNA FISH-FLOW except that cells were kept in suspension throughout the staining process. Briefly, cells were trypsinized to release them from monolayers, resuspended in 1% FBS/PBS, and fixed and permeabilized in 100% methanol for at least 15 min on ice. Hybridization was performed in 100 μl hybridization buffer containing 1.25 μM RNA FISH probes for 16 h in a humidified chamber at 37 °C. Cells were suspended in GLOX anti-fade media before flow cytometry/FACS. For live/dead staining, trypsinized cells were incubated with efluor-506 fixable viability dye (L/D Aqua, eBioscience) and diluted in 1% FBS/PBS in the presence of 40 U/ml RNase inhibitor (Invitrogen) prior to fixation. Data were acquired in a BD LSRFortessa instrument (filter 582/15-Green for the detection of Quasar 570-gSeV probe; 670/30-Red for the detection of Quasar 670-DVG probe; 515/30-Blue for the detection of L/D staining). A BD FACSAria II instrument was used for sorting RNA FISH hybridized cells. Sorted cells (≥90% purity from parental gate) were centrifuged and put in TRIzol (Invitrogen) for RNA extraction and RNA-Seq analysis. Flow cytometry data analysis was performed using the BD FACSDiva (BD Bioscience) or FlowJo (Tree Star) softwares.

**RNA-Seq of FISH FACS-sorted cells**. Four sub-populations of cells (ND, FL-high, DVG-high and INT) were sorted from infected cultures after RNA FISH. RNA-Seq was performed in two replicates where the RNA of each replicate was a pool from three independent FISH-FACS sortings (six independent sortings in total). RNA samples from FISH-FACS-sorted cells were prepared as follows: RNA extracted using TRIzol reagent was re-purified using the PicoPure RNA isolation kit (Thermo Fisher Scientific). Quality of the RNA was assessed by using the RNA Pico 6000 module on an Agilent Tapestation 2100 (Agilent Technologies). A schematic diagram for the FISH-FACS-sorting pipeline coupled with RNA-Seq is provided in Supplementary Fig. 5a. Briefly, total cDNA libraries were prepared starting from 75 ng of extracted raw RNA using the Illumina TruSeq Stranded Total RNA LT kit with Ribo-Zero Gold, according to the manufacturer's instructions. Samples were run on a Illumina NextSeq500 instrument to generate 75 bp, single-end reads, resulting in 21–53 million reads per sample with an average Q30 score ≥96.8%. All data were processed and analyzed using R programming language (v 3.2.2, R core team, 2016) and the RStudio interface (v 0.99.489), as described previously[46]. For host transcriptome analysis, raw fastq files were mapped to the human transcriptome (cDNA; Ensembl release 86) using Kallisto with 60 bootstraps per sample[46, 47]. Annotation and summarization of transcripts to genes was carried out in R, using the TxImport package[48]. Differentially expressed genes (≥twofold and ≤ 1% false discovery rate) were identified by linear modeling and Bayesian statistics using the VOOM function in Limma package[49, 50]. Gene Ontology (GO) was performed using the Database for Annotation, Visualization and Integration of Data (DAVID)[51]. GSEA was analyzed through the Molecular Signatures Database (MSigDB) using the C2 canonical pathway collection[52]. For genome alignment, seq reads from duplicates of each population were combined and aligned to the full length genome of SeV strain Cantell using the Subread aligner in the RSubread package[53]. This allowed us to extract the viral reads from the total. Resulting BAM files were visualized as a track on the full length SeV Cantell genome using the Genious software (version 7.1.9, Biomatters develop team) to obtain the different coverages of viral reads from each of the four populations.

**TNFR2 extracellular staining for flow cytometry**. Extracellular staining of TNFR2 on A549 cells for flow cytometry was performed as follows: cells harvested in 1% FBS/PBS were stained with anti-human-TNFR2 antibodies (2 μg/10^6 cells, R&D system, cat. no. AF726) in 1% FBS/PBS on ice for 40 min, washed three times with 1% FBS/PBS and then subjected to secondary antibody staining with goat anti-donkey Alexa Fluor 488 antibody (Invitrogen; 1:1000 diluted in 1% FBS PBS) on ice for 30 min. Stained cells were acquired in a BD LSRFortessa instrument and analyzed using the FlowJo (Tree Star) software.

**TNFα neutralization and supplementation of infected cells**. For TNFα neutralization experiments, 70–80% confluent A549 cells were pre-treated with neutralizing antibody against TNFα, TNFR1, and/or TNFR2 (all from R&D system, cat. nos. AF-210-NA, MAB225, and AF726, respectively) at 2 μg/ml/antibody or with IgG control antibody (R&D system, cat. no. MAB002) at the same concentration diluted in 400 μl of DMEM-2% FBS culture media. After 3–4 h, pre-treated cells were infected with SeV HD at a MOI of 1.5 diluted in 100 μl of the same media. Samples were collected after 20 and 30 h post infection for analysis. For long-term sub-culture, cells were treated with control IgG or TNFR2 neutralizing antibody on days 1, 3, and 5 post SeV HD infection and treated cells

were split and re-plated at the same cell number. Media was changed every 3–5 days depending on the condition of the cells. For TNFα supplementation experiments, human recombinant TNFα (Peprotech) was added to the media of infected cells at the indicated doses after 16 h of SeV HD infection. Samples were collected 8 h post treatment for further analysis.

**RNA interference**. siRNAs for human *TRAF1* (ON-TARGET plus smart pool, 7185, including 4 target sequences: GAAGGACGACACAAUGUUC, GAACU-CAGGAGAAGGCUCA, GGAAAGAGAACCCAUCUGU, UGUGGAAGAU-CACCAAUGU), human *BIRC3* (ON-TARGET plus smart pool, 330, including four target sequences: CAUGUGAACUGUACCGAAU, UAACGAAAAUGCCA-GAUUA, GUUCAUCCGUCAAGUUCAA, UCAACUGCCGGAAUUAUUA) and the ON-TARGET plus non-targeting control pool were obtained from GE health, Dharmacon. Briefly, $3 \times 10^4$ A549 cells were transfected with 50 μM of siRNAs using Lipofectamine RNAiMAX transfection reagent (Invitrogen) according to the manufacture's protocol. After 16 h of incubation, media was replaced with complete cell culture media without antibiotics. After 40 h of transfection, cells were infected with SeV HD (MOI = 1.5 TCID50 per cell) for 24 h. Cells were harvested with TRIzol for RNA preparations, put in P-40 lysis buffer for protein preparations, or fixed with 4% formaldehyde for image analysis. As control, cells were treated with equal amount of non-targeting scramble siRNA using Lipofectamine RNAi-MAX transfection reagent (Invitrogen).

**Statistical analysis**. All statistical analyses were performed with GraphPad Prism version 5.0 (GraphPad Software, San Diego, CA). A statistically significant difference was defined as a $P < 0.05$ by either one-way or two-way analysis of variance (ANOVA), significance of variance ($F$ value), or Student's $t$ with or without a post hoc test to correct for multiple comparisons on the basis of specific data sets as indicated in each figure legend.

**Data availability**. The RNA-Seq data that support the findings of this study have been deposited in Gene Expression Omnibus (GEO) database for public access (GSE96774). All other data are available upon request from the corresponding author.

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

## Acknowledgements

The authors wish to thank Alex Valenzuela for help with virus titrations. This work was supported by the US National Institutes of Health National Institute of Allergy and Infectious Diseases (NIH AI083284 and AI127832 to C.B.L. and AI104887 to S.W.) and The American Association of Immunologists Careers in Immunology Fellowship Program (AAI EIN 52-2317193 to C.B.L.). The PennVet Imaging Core Facility instrumentation is supported by NIH S10 RR027128.

## Author contributions

J.X., Y.S., and C.B.L. conceived experiments. J.X., Y.S., D.B., and A.R. developed methodology. J.X. and Y.S. performed experiments and collected data. G.R., Y.L., S.R.W., D.B., and A.R. provided reagents and resources. J.X. and C.B.L. wrote the original draft. C.B.L. supervised research activities.

## Additional information

**Competing interests:** The authors declare no competing financial interests.

