## [Peer Review File · Nature Communications]

Reviewers' comments:

Reviewer #1 (Remarks to the Author):

The primary results of this manuscript describe the interaction of defective virus genomes of Sendai virus with elements of the innate immune response that promote the establishment of long-lasting persistent infections in tissue culture cells. While the role of defective genomes in persistence has been known for some time and the stimulation of innate immune elements by defective interfering viruses has been described previously this is the first detailed examination of the interaction of the role of the defective genomes in persistence. The data are very clearly presented and demonstrate the involvement of the TNF pathway through TNFR2 and also of the mitochondrial antiviral-signalling protein in the promotion of persistence following infection with a virus preparation that contains a high level of defective virus genomes. These data are extremely interesting and the study was carried out in a very thorough manner with appropriate assays and confirmatory data using gene specific RT PCR to support genome wide transcriptional profiling and gene knockouts or specific antibodies to confirm specific gene requirements. All of these analyses are sound and the interpretation of the results is justified and measured. The data will undoubtedly be of interest and will lead to further discoveries about virus-host interactions.

However, while the main observations of the manuscript are well presented there are issues with some other parts of the manuscript which require attention to provide the necessary clarity and justification. Some of these issues may, in part, have arisen by the authors' seeking brevity but the result is in some areas may inadvertently lead to incorrect or incomplete understanding of the defective genome systems of viruses by readers for whom the data are of interest but who may not be fully versed in the field. In addition, there is limited justification provided for inclusion of some of the underpinning data and that should be addressed. It will be important to ensure that clarity is retained and the specific points for these two issues are given below.

In the abstract and elsewhere in the text the authors talk generically about defective virus genomes being involved in promoting or facilitating the establishment of persistent infections. While this is true it is important to point out that this feature has not been shown to be a property of all defective genomes. The authors do not anywhere point out that in the material that they have studied the defective genomes exist as a complex mixture of subgenomic molecules which, while they predominantly share the features that led to their description as copy back molecules, can differ widely from each other in the precise detail of the sequences they contain. Most importantly, not all defective genomes share the same biological properties, with examples from several systems, including Sendai virus, showing that two very similar molecules can differ significantly in the way that they interact with the host cell. With studies at the single cell level it is particularly important that this is clear as the differences in biological properties will affect to some degree the data that is obtained. It is not yet clear whether all defective genomes have the capacity to promote persistence but the language used by the authors is likely to be interpreted as indicating this is an established fact. Omitting these essential pieces of information may lead to misunderstanding of the real detail of the system being studied.

In a related issue, the authors refer in the first results section to the complementing of their material with 'purified defective particles containing DVGs'. No experimental detail is provided about the origin of these 'purified' particles or their production in the methods and a single reference in the discussion appears to be the only indication for their provenance. The use of the term 'purified' is not justified for this material as it is not possible to prepare paramyxovirus particles containing defective genomes in the absence of particles containing full length genomes. The most appropriate description would be 'enriched' and the authors should also clearly indicate whether/how the infectious virus was inactivated. If possible it would be useful to know what degree of enrichment was achieved.

The initial data presented in the manuscript describes the single cell assay that the authors have developed and used in the subsequent analysis. The inclusion of this is important and justified but it requires more detail in the methods section. In particular, the fluorescence analysis clearly shows the presence of cells that contain predominantly defective genomes and others that contain predominantly full length genomes. An additional fraction contain significant levels of both types of genomes but these are not pursued further in this study. There is no indication of the sensitivity of the fluorescence assay to detect the two types of RNA and this raises some questions. If the authors have some assessment of sensitivity it would be very useful to include it e.g. is one type of molecule more easily detected in this system than the other? Assuming that the sensitivity of detection is not significantly different for the two types of molecule the result begs the question of what the true multiplicity of infection was in the cultures. The authors quote an m.o.i. based on TCID50 but the most important issue of the precise m.o.i. of infectious virus is not clear from this. The data would suggest that there was only sufficient infectious virus to infect a proportion of the cells which, if correct would explain why some cells contain only defective genomes as particles containing these vastly outnumber the particles containing the full length genomes in the appropriate virus stock. The presence of cells that were negative for both types of virus RNA also implies that the m.o.i. may have been low. If this is the case the observation is not surprising and does not require a detailed consideration, though the inclusion of the images and the analyses remain justified for the manuscript. The inclusion of the image in Figure 1E showing detection of full length and defective RNA in cells from the mouse respiratory tract after infection with a stock of Sendai virus containing high levels of defective genomes serves no real purpose other than to show that defective virus particles can deliver their genome cargo to susceptible cells. This has been known for some time and particularly given that the manuscript does not contain any other in vivo-derived data it should be removed.

Figure 3 contains a similar single cell analysis following infection with respiratory syncytial virus stocks containing relatively high or low levels of defective genomes. This is presented to demonstrate that a different virus generates a similar picture to that seen with Sendai virus. These data do not contribute to any of the conclusions of the study and its inclusion is not necessary, particularly as RSV is genetically so similar to Sendai virus, having until recently been classed within the same virus family. No further analysis of RSV is shown despite the discussion referring to RSV-induced apoptosis being reduced in MAVs KO cells. Removal of the RSV data would not impair the manuscript and would enhance focus on the

main virus system that is explored in detail.

Overall, the manuscript provides convincing data about the role defective virus genomes interacting with host factors to promote persistent infection. It contains new data on an interesting and fundamental aspect of virus-host interaction that identifies host factors that are key players in the process and does so in a convincing and clear manner.

Reviewer #2 (Remarks to the Author):

Xu and colleagues developed RNA FISH assay to analyze the mechanisms of replication defective viral genomes (DVGs) associated persistent Sendai virus and respiratory syncytial virus infection. This assay is central to the analysis and the authors carefully characterize its sensitivity including mutant genomes that cannot form DVGs. To quantify DVGs in infected cells Xu et al use fluorescence microscopy and - similar to previous RNA flow based methods - flow cytometry. Using this new RNA FISH based method which can distinguish between DVGs and replicating genomes the authors show a considerable heterogeneity in DVG quantities across populations of virally infected cells in culture and SeV-infected mice. The authors observed that the DVG-high cells survive the infection longer than cells enriched in full-length virus, establishing the persistent infection. The survival of DVG-high cells is dependent on MAVS/TNFA/TNFR2 axis, which is demonstrated by knockout/knockdown or neutralizing antibodies blocking assays.

Overall, this is a well designed and conducted study provides into mechanisms by which distinct viral genomic products (of certain viruses) influence cell fate upon infection and also reveals the dual functions of TNFA in the viral infection to perpetuate host and virus. However, besides some technical issues outlined below, a concern is how generalizable these concepts are and whether the cell culture phenomena hold up in vivo. The authors confirm that DVG are heterogeneously distributed following SeV infections in mice but all of the other data are based on use of cell lines. Arguably the greatest weakness of this study is whether the higher numbers of DVG are causal to the persistence phenotype. The data provided here are suggestive of but do not prove a causal relationship.

Specific comments

1. The author should make it clearer throughout the manuscript including the abstract that they are looking at two specific viruses (SeV and RSV) and that mechanisms for viral persistence could and in fact are quite different for other viruses.
2. The authors used CRISPR/Cas9 to knockout MAVS, aiming to test the role of MAVS in DVG associated cell survival. The authors just used one set of gRNA to knockout MAVS but it would be advisable to repeat this assay with other independent sgRNA to rule out off-target issues. More importantly the author would have to perform rescue assay to confirm the specificity, which will make this point more convincing.
3. The authors proposed that TNFR2 is important for the DVG-high cells to survive (Fig 6A and B). The authors just used the neutralizing antibody to block the TNFR2 to demonstrate this point. The authors should consider the genetic method, such as CRISPR/Cas9 to test TNFR2 function in this system. It will be more interesting if the author could switch the TNFR1 and

TNFR2 expression in FL-gSeV cells and DVG-high cells to measure the cell survival.

4. Page 6, first paragraph. The authors could add a bit more description in the text how the in vivo experiments were conducted (this reviewer is aware that the details are in the M&Ms)

5. A statement should be included in the M&Ms (mouse experiments) that all animal experiment were performed under IACUC approved protocols (number?)

Reviewer #3 (Remarks to the Author):

The authors address an important question regarding the role of defective viral RNAs (DVGs) in paramyxovirus persistence using imaging probes that distinguish between DVGs and standard viral genomes. These studies showed that DVGs accumulate in a subset of infected cells that survive longer than infected cells enriched in full-length genomes. Furthermore, they identify the mechanism as a MAVS-dependent resistance to TNF-mediated apoptosis. Attention to the following would improve the manuscript:

1. Intro, pg 3, para 2 – the innate response controls infection, but clearance usually requires the adaptive immune response.
2. Results, pg 5 – How are pDPs prepared?
3. Methods, pg18, viruses, ln 4 – What is the infectious: total particle ratio of SeV Cantell LD?
4. Fig 1 legend and Methods, pg18, mice – which stock of SeV, LD or HD was used for infection?
5. Discussion - It is not clear what drives the heterogeneity of the response – some speculation as to mechanism and whether this occurs in vivo as well as in vitro should be included.

Minor issues

1. Intro, pg 3, para 2, ln 7 – replication of defective . . .
2. Results, pg 7, para 2, ln 1 – “impacts”
3. Results, pg 8, para 1, ln 3 – I assume this is cleaved PARP-1. If so, it should be identified as such.
4. Results, pg 11, ln 15 – “impacts”
5. Results, pg 11, ln 17-19 – awkward sentence, maybe: “while it increased apoptosis . . . , suggesting that TNFa signaling protected DVG-high cells and killed . . .”
6. Methods, pg 18, viruses, Ln 9 “medium or median”?

Point by Point response:

First, we want to thank the reviewers for their positive and helpful comments on our original submission. We have responded to all the suggestions and commentaries, and this has certainly resulted in an improved manuscript. We have marked in **burgundy** font those sections in the manuscript that directly answer to the reviewers' concerns. Below is a detailed point by point response:

Reviewer #1:

The primary results of this manuscript describe the interaction of defective virus genomes of Sendai virus with elements of the immune response that promote the establishment of long-lasting persistent infections in tissue culture cells. While the role of defective genomes in persistence has been known for some time and the stimulation of innate immune elements by defective interfering viruses has been described previously this is the first detailed examination of the interaction of the role of the defective genomes in persistence. The data are very clearly presented and demonstrate the involvement of the TNF pathway through TNFR2 and also of the mitochondrial antiviral-signaling protein in the promotion of persistence following infection with a virus preparation that contains a high level of defective virus genomes. These data are extremely interesting and the study was carried out in a very thorough manner with appropriate assays and confirmatory data using gene specific RT PCR to support genome wide transcriptional profiling and gene knockouts or specific antibodies to confirm specific gene requirements. All of these analyses are sound and the interpretation of the results is justified and measured. The data will undoubtedly be of interest and will lead to further discoveries about virus-host interactions.

R: We thank the reviewer for this positive assessment.

*However, while the main observations of the manuscript are well presented there are issues with some other parts of the manuscript which require attention to provide the necessary clarity and justification. Some of these issues may, in part, have arisen by the authors' seeking brevity but the result is in some areas may inadvertently lead to incorrect or incomplete understanding of the defective genome systems of viruses by readers for whom the data are of interest but who may not be fully versed in the field. In addition, there is limited justification provided for inclusion of some of the underpinning data and that should be addressed. It will be important to ensure that clarity is retained and the specific points for **these two issues are given below.***

In the abstract and elsewhere in the text the authors talk generically about defective virus genomes being involved in promoting or facilitating the establishment of persistent infections. While this is true it is important to point out that this feature has not been shown to be a property of all defective genomes. The authors do not anywhere point out that in the material that they have studied the defective genomes exist as a complex mixture of subgenomic molecules which, while they predominantly share the features that led to their description as copy back molecules, can differ widely from each other in the precise detail of the sequences they contain. Most importantly, not all defective genomes share the same biological properties, with examples from several systems, including Sendai virus, showing that two very similar molecules can differ significantly in the way that they interact with the host cell. With studies at the single cell level it is particularly important that this is clear as the differences in biological properties will affect to some degree the data that is obtained. It is not yet clear whether all defective genomes have the capacity to promote persistence but the language used by the authors is likely to be interpreted as indicating this is an established fact. Omitting these essential pieces of information may lead to misunderstanding of the real detail of the system being studied.

R: We thank the reviewer for this comment that helps improving the clarity and accuracy of our report. We agree that, in general, it is likely that distinct DVG species exist in the cultures and that their activity may vary. We have modified the abstract and introduction to reflect this possibility, while responding to the journal's word count limitations. We also want to emphasize that our RNA-FISH and conclusions are strictly referring to the "copy-back" type of DVGs

formed during single strand negative sense virus replication. This is an important distinction since SeV strain Cantell has one predominant well-characterized copy-back DVG that we have confirmed as the largely predominant copy-back DVG (shown in Fig. S4D using RNA-Seq data, as well as in our previous publications analyzing copy-back DVGs by PCR (Tapia et al, PLOS Pathogens, 2013)) and in publications by the Kolakofsky group. RSV, in contrast, produces a discrete population of copy-back DVGs (described in detail in our publication Sun, et al, PLOS Pathogens, 2015), and our probes are designed to capture the large majority of these copy back DVGs as explained in the manuscript. We have provided additional clarity of these facts by including text expanding on this issue in the discussion. This text reads as follows:

“Given the diversity of DVGs, it is likely that distinct DVG species are generated during infection and that the ability of different DVGs to promote virus persistence varies. Our study assessed exclusively the function of copy-back DVGs. As copy-back DVGs confer the most potent immunostimulatory activity to paramyxovirus infections it is relevant to follow their activity, regardless of the potential function of other DVGs. Moreover, we provide evidence that the predominant copy-back DVG-546 species present in cultures infected with SeV HD^{9,22,31} is the predominant copy-back species our assays (Fig. S4D). RSVs, in contrast, produces a discrete population of copy-back DVGs³² and our probes are designed to capture the large majority of these copy back DVGs”

In a related issue, the authors refer in the first results section to the complementing of their material with ‘purified defective particles containing DVGs’. No experimental detail is provided about the origin of these ‘purified’ particles or their production in the methods and a single reference in the discussion appears to be the only indication for their provenance. The use of the term ‘purified’ is not justified for this material as it is not possible to prepare paramyxovirus particles containing defective genomes in the absence of particles containing full length genomes. The most appropriate description would be ‘enriched’ and the authors should also clearly indicate whether/how the infectious virus was inactivated. If possible it would be useful to know what degree of enrichment was achieved.

R: We thank the reviewer for noticing our failure to include the DP purification procedure. We also apologize for any misunderstanding that this oversight might have caused. An explanation of the pDPs preparation is now included in the methods section of the manuscript under the heading “Purified defective particles (pDPs) preparation”, along with descriptions of quality control showing degree of defective particle enrichment. As further explanation, we want to highlight that the pDP used in Fig.1a and the new **Supplementary Fig. 2b-f** of this study are actually purified through two rounds of density gradient centrifugation and separation. Paramyxovirus DPs have lower density than virions containing full-length genomes and we and others have extensively used this method to purify DPs in the past. As suggested by the extremely low TCID₅₀/HA ratio (96.9), and the inability of this fraction to replicate in cells in the absence of co-infecting standard virus, this viral fraction is highly purified. We want to note that for the rest of the manuscript, we use viruses expanded to contain a High or Low DVG content as indicated throughout. We have now included the TCID₅₀/HA ratio for these viruses in the methods section as a comparative reference (HD TCID₅₀/HA=98120, LD TCID₅₀/HA =4916). The new methods section read as follows:

“Purified defective particles (pDPs) preparation.

pDPs preparation was performed as previously described¹⁰. Briefly, allantoic fluid from 100 SeV Cantell-infected embryonated eggs was concentrated by high-speed centrifugation. The resulting pellets were suspended in 0.5 ml of PBS/2 mM EDTA and incubated overnight at 4°C. The suspension was then added on top of a 5–45% sucrose (Fisher) gradients prepared using a gradient maker (BioComp). Gradients were centrifuged at 28,000 rpm for 1.5 h and the fraction containing low-density viral particles was collected and re-purified using the same procedure. Final low-density fractions were concentrated by centrifugation at 21,000 rpm for 2 h. Pellets were suspended in PBS and stored at –80°C. The content of pDPs particles was determined by calculating the TCID₅₀/HA ratio as reflection of infectious over non-infectious particles as described previously¹⁰. pDPs used in this study had a TCID₅₀/HA ratio of 96.9”

The initial data presented in the manuscript describes the single cell assay that the authors have developed and used in the subsequent analysis. The inclusion of this is important and justified but it requires more detail in the methods section. In particular, the fluorescence analysis clearly shows the presence of cells that contain predominantly defective genomes and others that contain predominantly full length genomes. An additional fraction contains significant levels of both types of genomes but these are not pursued further in this study. There is no indication of the sensitivity of the fluorescence assay to detect the two types of RNA and this raises some questions. If the authors have some assessment of sensitivity it would be very useful to include it e.g. is one type of molecule more easily detected in this system than the other?

R: As requested, we have now provided a comparison of DVG and FL probes sensitivity based on copy number quantitation of (+)DVG and (+)FL-gSeV. New data are shown in **Supplementary Fig. 1f**. Data show that sensitivity of detection of DVG and FL genomes based on RNA-FISH and imaging corresponds exactly with sensitivity of detection of these viral genome species by RT-qPCR. New text reads:

“Importantly, both FL-gSeV and DVGs probe pools have comparable sensitivity and similar sensitivity to viral product-specific RT-qPCR (Supplementary Fig. 1f)”.

Assuming that the sensitivity of detection is not significantly different for the two types of molecule the result begs the question of what the true multiplicity of infection was in the cultures. The authors quote an m.o.i. based on TCID50 but the most important issue of the precise m.o.i. of infectious virus is not clear from this. The data would suggest that there was only sufficient infectious virus to infect a proportion of the cells which, if correct would explain why some cells contain only defective genomes as particles containing these vastly outnumber the particles containing the full length genomes in the appropriate virus stock. The presence of cells that were negative for both types of virus RNA also implies that the m.o.i. may have been low. If this is the case the observation is not surprising and does not require a detailed consideration, though the inclusion of the images and the analyses remain justified for the manuscript.

R: We understand the reviewer’s concern in regard of the apparent low moi of infection in some of the data presented, and would like to address this concern in the following two points:

1) The TCID50 titer of SeV is calculated on the highly susceptible cell line LLC-MK2. Data in Fig. 1d and Fig. 2 correspond to infections in this cell line and, as expected, roughly 80-90% of the cells show viral signal when using RNA-FISH for detection. We agree that this moi is not equivalent when using less susceptible cells, such as A549. We have done two things to address this issue in the manuscript:

a) We have further specified in the methods section that the moi used throughout the manuscript refers to that on LLC-MK2 cells used for titration (new text reads: *“Of note, to maintain consistency, the MOI used was calculated throughout the manuscript based on the titration of virus stocks in LLC-MK2 cells, regardless of the cell type infected”*).

b) We have included A549 infection at a higher moi showing that differential accumulation of DVG and FL-gSeV among cells is still observed (**new Figs 1e and Supplementary Fig.1e**).

2) Analysis of infected populations using RNA-FISH-flow in A549 cells may give the erroneous impression of an even lower percentage of cells infected. We want to clarify that the “ND” gate corresponds to “non-detected” not to “non-infected”. As shown in our RNA-seq analysis (Supplementary Fig. 4d), there is a substantial number of reads aligning to the viral genome in cells that fall into the ND gate, indicating that cells in this population are infected but containing very low amount of viruses. To further bring clarity to this point, we have now included the number of reads that aligned to the genome from each subpopulation in Supplementary Fig. 4d.

The inclusion of the image in Figure 1E showing detection of full length and defective RNA in cells from the mouse respiratory tract after infection with a stock of Sendai virus containing high levels of defective genomes serves no real purpose other than to show that defective virus particles can deliver their genome cargo to susceptible cells. This has been known for some time and particularly given that the manuscript does not contain any other in vivo-derived data it should be removed. Figure 3 contains a similar single cell analysis following infection with respiratory syncytial virus

stocks containing relatively high or low levels of defective genomes. This is presented to demonstrate that a different virus generates a similar picture to that seen with Sendai virus. These data do not contribute to any of the conclusions of the study and its inclusion is not necessary, particularly as RSV is genetically so similar to Sendai virus, having until recently been classed within the same virus family. No further analysis of RSV is shown despite the discussion referring to RSV-induced apoptosis being reduced in MAVs KO cells. Removal of the RSV data would not impair the manuscript and would enhance focus on the main virus system that is explored in detail.

R: We agree with the reviewer that removing the in vivo data in Fig. 1 makes sense as all other data is in vitro, thus, we have removed it as suggested. In regards to the RSV data, after consultation with the editors we have decided to keep these data in the manuscript. Our rationale is that these data demonstrate that viral persistence and differential cell death of DVG high and low cells is not only a “SeV effect”, but is also observed with the same methodology during infection with a different, and clinically relevant, paramyxovirus. We believe these data speak to the generalization of the observations that motivate the following mechanistic studies.

Reviewer #2:

Overall, this is a well designed and conducted study provides into mechanisms by which distinct viral genomic products (of certain viruses) influence cell fate upon infection and also reveals the dual functions of TNF α in the viral infection to perpetuate host and virus. However, besides some technical issues outlined below, a concern is how generalizable these concepts are and whether the cell culture phenomena hold up in vivo. The author confirm that DVG are heterogeneously distributed following SeV infections in mice but all of the other data are based on use of cell lines

R: We agree with the reviewer that demonstration that the phenomenon holds up in vivo would be optimal. In fact, we are actively working in generating tools that would allow us to address this question. Regrettably, at the current state of the field there are important limitations that preclude a robust answer. Below some of these issues and the approaches that we have taken:

- 1) All paramyxoviruses generate DVGs during their replication in vivo. Therefore, we don't have a natural system to compare DVG+ and DVG- viruses.
- 2) The mechanisms modulating the generation of copy-back DVGs during paramyxovirus replication are unknown and in new evidence from our lab, this mechanism is distinct from that modulating the generation of quasispecies (polymerase activity). Therefore, we currently can't generate recombinant viruses to compare in vivo until the molecular mechanisms are resolved.
- 3) We have tried multiple approaches to address the role of DVGs in promoting viral persistence in vivo. These approaches included comparing infections with HD and LD viruses and adding purified defective particles to LD infections. Regrettably, although the persistence of SeV genomes in the lung is observed 3 months after infection, LD viruses can generate DVGs during infection and also persist. In addition HD viruses (or viruses supplemented with purified DVGs) replicate to lower titers than LD viruses due to DVG-induced antiviral activity, making comparisons among long term HD and LD infections hard to interpret. The direct contribution of DVGs to persistency in vivo is therefore hard to prove in this system.
- 4) We've infected IFNRKO mice to minimize the difference in growth between LD and HD viruses in mice. In this scenario, both viruses grow to higher levels and produce more DVGs!...the level of persistence in this situation is higher than in WT mice, serving as an argument in favor of a role for DVGs, but I consider this interpretation a stretch and not a direct prove.

In conclusion, although we have suggestive data, demonstration of the role of DVGs in promoting persistence *in vivo* is currently not possible. This issue has become a priority of my laboratory and we are committed to demonstrate this in the future.

Arguably the greatest weakness of this study is whether the higher numbers of DVG are causal to the persistence phenotype. The data provided here are suggestive of but do not prove a causal relationship.

R: To address this concern, we have now included new data that more directly addresses the role of DVGs in persistence. In this new Figure (**Supplementary Fig. 2b-e**) we show that

supplementation of SeV LD infection with purified defective viral particles leads to cell survival and persistence, while supplementation with UV-inactivated purified defective viral particles doesn't. We believe that these data supplement the LD/HD virus experiments and strengthen the argument of a causal role for DVGs in promoting viral persistence. The new text reads:

“A direct role for DVGs in promoting the generation of persistently infected cultures was confirmed using pDPs to supplement SeV LD infections. As expected, extended survival of the infected population was rescued by supplementation with pDPs whereas supplementation with UV inactivated pDPs was not able to promote cell survival (Supplementary Fig. 2b,c). Similarly to SeV HD infected cells, the survivors from infection of SeV LD supplemented with pDPs contain highly replicative viral genomes and retained a significant population of DVG-high cells (Supplementary Fig. 2d-f).”

Specific comments

1. The author should make it clearer throughout the manuscript including the abstract that they are looking at two specific viruses (SeV and RSV) and that mechanism for viral persistence could and in fact are quite different for other viruses.

R: We have adjusted the text in the abstract, introduction, and discussion to make these points clearer. The modified abstract and intro texts are included below:

Abstract: *“...We report that during Sendai and respiratory syncytial virus infections DVGs selectively protect a subpopulation of cells from death and promote the establishment of persistent infections. We find that during Sendai virus infection this phenotype results from DVGs stimulating a MAVS-mediated TNF response that drives apoptosis of highly infected cells while extending the survival of cells enriched in DVGs...”*

Introduction: *“Using fluorescent in situ hybridization targeting ribonucleic acid molecules (RNA FISH) to distinguish DVGs from standard viral genomes during infection, we reveal that during infection with the murine parainfluenza virus Sendai (SeV) or RSV DVGs accumulate only in a subpopulation of infected cells, and that these cells survive the infection longer than cells enriched in full-length virus, leading to the establishment of persistent infections. Moreover, the survival of DVG-high cells is dependent on MAVS signaling, and we identify TNF α produced in response to MAVS signaling as pivotal in determining cell fate during SeV infection”*.

2. The authors used CRISPR/Cas9 to knockout MAVS, aiming to test the role of MAVS in DVG associated cell survival. The authors just used one set of gRNA to knockout MAVS but it would be advisable to repeat this assay with other independent sgRNA to rule out off-target issues. More importantly the author would have to perform rescue assay to confirm the specificity, which will make this point more convincing.

R: We understand the reviewer's concern with potential off-targets issues in our CRISPR MAVS knocked out cells. Following a bullet point response explaining how we have assessed the quality of these cells. In addition, these cells have now been published (Li et al, eLife, 2017).

- 1) The assays were repeated with a completely independently generated CRISPR generated MAVS KO cells using a different sgRNA. As we failed to described this is the original version, we have now included this information in the materials and methods section.
- 2) We performed a complementation assay as requested by transfecting a MAVS expression plasmid into CRISPR MAVS KO cell lines. We have now included these data in **Supplementary Fig. 5a**. The new text reads as follows:

“Specific elimination of MAVS activity was confirmed upon transfection of $0.25\mu\text{g}/10^5$ cells of MAVS-expression plasmid (MAVS-WT, Addgene) into KO cells and measuring antiviral gene expression 16 h post infection (Supplementary Fig. 5a). In addition, most experiments were repeated in a second independently generated MAVS KO cell line that used”

3. The authors proposed that TNFR2 is important for the DVG-high cells to survive (Fig 6A and B). The authors just used the neutralizing antibody block the TNFR2 to demonstrate this point. The authors should consider the genetic method, such as CRISPR/Cas9 to test TNFR2 function in this system. It will be more interesting if the author could switch the TNFR1 and TNFR2 expression in FL-gSeV cells and DVG-high cells to measure the cell survival.

R: We considered CRISPR KO of TNFR2, but concluded that this was unlikely to give us a robust phenotype since the activity of TNFR2 can be compensated by some of the other many TNFR2-like TNFRs expressed in the cells. For this reason, instead of going for TNFR2 directly for validation of the pathway, we also knocked down TRAF1, a critical signaling molecule for all anti-apoptotic TNFRs. Data on this KDs (Figs. 6c and d) strongly recapitulates neutralization of TNFR2. Regarding switching receptor expression in the populations, regrettably this is currently technically impossible, as we can only distinguish the populations after fixing the cells and performing RNA-FISH, which means the cells are dead and already committed to either fate.

In order to respond to the reviewer's concern and further prove the role of TNFR2 in promoting virus persistence, we now provide two additional pieces of data:

- 1) Knock down of BIRC3, a pro-survival gene regulated by TNFR2, shows impaired survival of DVG-high cells (new **Supplementary Fig 6b-e**). New text reads:

"In agreement with a pro-survival role for TNFR2 in DVG-high cells, knockdown of the TNFR2 adaptor molecule TRAF1 or the downstream effector BIRC3 increased apoptosis in DVG-high but not in FL-high cells (Fig. 6c,d and Supplementary Fig. 6)".

- 2) Long term culture of SeV HD infected cells treated with TNFR2 neutralizing antibodies demonstrate that blocking this receptor impairs the generation of persistent infections. In this experiment, A549 cells infected with SeV HD and treated with TNFR2 neutralizing antibodies at early times post infection were maintained in culture for up to 28 days before analysis. As shown in new **Fig. 6g-j**, limiting TNFR2 engagement impaired the long term survival of infected cells. The resulting cell cultures, although recovered at later time points, contained only ~5% of SeV infected cells as compared to ~45% seen in control SeV infected long-term cultures. New text reads:

"Furthermore, treatment of SeV HD infected cells with TNFR2 neutralizing antibody leads to a crisis of massive cell death, similar to infections with SeV LD, and cells that eventually recover show a significantly reduced percentage of infection after long-term subculture compared to untreated cells (Fig. 6g-j)."

4. Page 6, first paragraph. The authors could add a bit more description in the text how the *in vivo* experiments were conducted (this reviewer is aware that the details are in the M&Ms)

R: As we have removed the *in vivo* experiment from the manuscript in response to a request to aid focus from reviewer 1, this text is not longer included.

5. A statement should be included in the M&Ms (mouse experiments) that all animal experiment were performed under IACUC approved protocols (number?)

R: Please refer to the "ethics statement" section in the material and methods section in the initial submission for the requested information. However, as we removed the *in vivo* experiment completely, that information is excluded from the updated version of the manuscript.

Reviewer #3:

The authors address an important question regarding the role of defective viral RNAs (DVGs) in paramyxovirus persistence using imaging probes that distinguish between DVGs and standard viral genomes. These studies showed that DVGs accumulate in a subset of infected cells that survive longer than infected cells enriched in full-length genomes. Furthermore, they identify the mechanism as a MAVS-dependent resistance to TNF-mediated apoptosis. Attention to the following would improve the manuscript:

1. Intro, pg 3, para 2 – the innate response controls infection, but clearance usually requires the adaptive immune response.

R. As suggested, we have revised the text in the introduction to improve clarity. Paragraph 2 of the intro now reads:

“The innate immune response is the first active host barrier to virus replication and is essential to control the infection and activate adaptive responses that result in virus clearance”.

2. Results, pg 5 – How are pDPs prepared?

R. We thank the reviewer for noticing our failure to include the DP purification procedure. We also apologize for any misunderstanding that this oversight might have caused. An explanation of the pDPs preparation is now included in the methods section of the manuscript under the heading “Purified defective particles (pDPs) preparation”, along with descriptions of quality control showing degree of defective particle enrichment. The modified text is included in response to a similar concern from Reviewer 1.

3. Methods, pg18, viruses, In 4 – What is the infectious: total particle ratio of SeV Cantell LD?

R. The SeV Cantell LD we used in this study has an infectious: total particle ratio of 98120 (infectivity of \log_{10} 7.4 TCID₅₀/25ul and direct HA of 256/25ul). We have also updated this description into the methods section under the heading “Viruses” and included the TCID₅₀/HA ratios for all our preparations. The text reads:

“SeV LD used in this study has an infectious: total particle ratio (TCID₅₀/HA titer) of 98120, SeV HD used in this study has an infectious: total particle ratio (TCID₅₀/HA titer) of 4916”.

4. Fig 1 legend and Methods, pg18, mice – which stock of SeV, LD or HD was used for infection?

R. We apologize for this oversight. Although these data were removed from the manuscript in response to Reviewer’s 1 suggestion, the infection was performed with SeV 52. This is a LD strain that naturally generates DVGs during viral replication [Tapia et.,al Plos Pathogen 2013].

5. Discussion - It is not clear what drives the heterogeneity of the response – some speculation as to mechanism and whether this is occurs in vivo as well as in vitro should be included.

R. In response to this request, we have now extended the discussion and included a new second to the last paragraph on this topic. The paragraph reads:

“The mechanism driving the heterogeneity of FL genomes and DVG distribution and associated innate immune responses during initial viral infection remains unclear. Cell to cell variation in type I IFN expression has been demonstrated in a number of systems in association with differential expression of innate immune limiting factors^{44, 45}. These studies implied that selected host factors may control differential responses towards virus infection. Interestingly, our data show that the heterogeneous distribution of FL genomes and DVGs among infected cells along with the associated differential cell death rate is not significantly altered in IFNAR1 KO or MAVS KO cells (Figure 5A), suggesting that factors additional to the type I IFN pathway are involved. One possibility is that the cell cycle status at the time of infection is a critical factor in determining heterogeneity. Another possibility is that stochastic DVG accumulation in infected cells drives the phenotype. In this case, it is expected that DVG accumulation is a dominant phenotype and when cells are seeded with DVGs, either through stochastic infection with viral particles containing DVGs, or through generation of DVGs during virus replication, DVGs take over the viral replication process. Interestingly, experiments not shown show that natural accumulation of DVGs during SeV replication in vivo follows a similar heterogenic pattern, where some cells in the lung epithelium accumulate DVGs and others don’t, demonstrating that this process is not limited to in vitro infections”.

Minor issues

1. *Intro, pg 3, para 2, Ln 7 – replication of defective . . .*
2. *Results, pg 7, para 2, Ln 1 – “impacts”*
3. *Results, pg 8, para 1, Ln 3 – I assume this is cleaved PARP-1. If so, it should be identified as such.*
4. *Results, pg 11, Ln 15 – “impacts”*
5. *Results, pg 11, Ln 17-19 – awkward sentence, maybe: “while it increased apoptosis . . . , suggesting that TNFa signaling protected DVG-high cells and killed . . .”*
6. *Methods, pg 18, viruses, Ln 9 “medium or median”?*

R: We thank the reviewer for noting these mistakes. They have been corrected in the text with the following exceptions:

Point 1 is correct as stated. Adding “of” incorrectly changes the meaning of the sentence. For point 6, medium is correct as per convention.

REVIEWERS' COMMENTS:

Reviewer #1 (Remarks to the Author):

The authors have addressed the points raised in the reviews and the alterations to the text and the additional information has clarified the issues raised. On the question of the multiplicity of infection the authors have gone some way to addressing the point by indicating that the level of infectivity in the two cell lines is different which means that using the same inocula is not entirely appropriate. Using a higher level of infectivity with the less permissive cells does go some way to dealing with this. However, the main point being raised was that with a multiplicity of infection that does not provide sufficient infectious virus to deliver a fully infectious genome to every cell it is not surprising that there will be some cells that contain only a defective genome. Thus, the observation that a culture contained cells with only a defective genome does not merit a great deal of consideration - though the implication that these cells would then be likely to provide an environment that facilitates persistence does. The authors' response that 'none detected' does not demonstrate absence is completely reasonable but was not the aim of the original comment. In reality this is a minor issue but clarity is important and only a minor alteration to indicate that a much higher level of infectivity to introduce fully infectious virus into every cell may potentially alter the outcome in terms of the extent of persistence established.

Reviewer #2 (Remarks to the Author):

The authors have prepared a point-by-point rebuttal and included additional data which carefully address the points that I have raised earlier. This reviewer recognizes that due to current technical limitations in the field. The authors should consider adjusting the title to "Replication defective viral genomes exploit a cellular pro-survival mechanism to establish RSV and Sendai virus persistence".

Reviewer #3 (Remarks to the Author):

The authors have adequately addressed my issues.

Point by Point:

Reviewer #1 (Remarks to the Author):

The authors have addressed the points raised in the reviews and the alterations to the text and the additional information has clarified the issues raised. On the question of the multiplicity of infection the authors have gone some way to addressing the point by indicating that the level of infectivity in the two cell lines is different which means that using the same inocula is not entirely appropriate. Using a higher level of infectivity with the less permissive cells does go some way to dealing with this. However, the main point being raised was that with a multiplicity of infection that does not provide sufficient infectious virus to deliver a fully infectious genome to every cell it is not surprising that there will be some cells that contain only a defective genome. Thus, the observation that a culture contained cells with only a defective genome does not merit a great deal of consideration - though the implication that these cells would then be likely to provide an environment that facilitates persistence does. The authors' response that 'none detected' does not demonstrate absence is completely reasonable but was not the aim of the original comment. In reality this is a minor issue but clarity is important and only a minor alteration to indicate that a much higher level of infectivity to introduce fully infectious virus into every cell may potentially alter the outcome in terms of the extent of persistence established.

R: We would like to clarify the characteristics of the DVG-high cell population, as we believe that a misunderstanding is at the core of the reviewer's comment. By definition, copy-back defective viral genomes (DVGs) are replication incompetent and therefore require full-length viral genome (FL) for further propagation once they are generated. Thus, each DVG-high cell has to contain at least one copy of full infectious viral genomes to provide enough viral replication machinery for DVGs to replicate and amplify. Our data support this statement. qPCR and RNA-seq data of the three sorted populations (ND, DVG-high, and FL-high) in Fig 1h and supplementary Fig 4d show that the DVG-high population contains a significant amount of full-length viral genome that exceeds the amount contained in the ND population, but is less than that in FL-high cells. Thus, DVG-high cells are highly enriched in DVGs over full-length viral genomes, but they are not depleted of full-length viral genomes. To further clarify this point, we have added the following sentence to the results section:

"Note that DVG-high cells also contain a relatively small amount of full-length viral genomes, which is expected to provide the viral machinery for DVG replication".

Reviewer #2 (Remarks to the Author):

The authors have prepared a point-by-point rebuttal and included additional data which carefully address the points that I have raised earlier. This reviewer recognizes that due to current technical limitations in the field. The authors should consider adjusting the title to "Replication defective viral genomes exploit a cellular pro-survival mechanism to establish RSV and Sendai virus persistence".

R: We appreciate reviewer #2 comment on the title. Since RSV and Sendai virus belong to two distinct genera within the paramyxoviruses, we decided to modify the title as follows: "Replication defective genomes exploit a cellular pro-survival mechanism to establish paramyxovirus persistence".

Reviewer #3 (Remarks to the Author):

The authors have adequately addressed my issues.

R: We thank again reviewer #3 for carefully looking at our manuscript.